# Unearthing Skill-Level Insights for Understanding Trade-Offs of Foundation Models

**Mazda Moayeri**[1,2*], **Vidhisha Balachandran**[1], **Varun Chandrasekaran**[1,3],

**Safoora Yousefi**[1], **Thomas Fel**[4], **Soheil Feizi**[2], **Besmira Nushi**[1], **Neel Joshi**[1], **Vibhav Vineet**[1]

[1]Microsoft Research AI Frontiers, [2]University of Maryland,
[3]University of Illinois Urbana-Champaign, [4]Harvard University Kempner Institute
mmoayeri@umd.edu, vivineet@microsoft.com

## Abstract

With models getting stronger, evaluations have grown more complex, testing multiple skills in one benchmark and even in the same instance at once. However, skill-wise performance is obscured when inspecting aggregate accuracy, under-utilizing the rich signal modern benchmarks contain. We propose an automatic approach to recover the underlying skills relevant for any evaluation instance, by way of inspecting model-generated *rationales*. After validating the relevance of rationale-parsed skills and inferring skills for 46k instances over 12 benchmarks, we observe many skills to be common across benchmarks, resulting in the curation of hundreds of *skill-slices* (i.e. sets of instances testing a common skill). Inspecting accuracy over these slices yields novel insights on model trade-offs: e.g., compared to GPT-4o and Claude 3.5 Sonnet, on average, Gemini 1.5 Pro is $18\%$ more accurate in *computing molar mass*, but $19\%$ less accurate in *applying constitutional law*, despite the overall accuracies of the three models differing by a mere $0.4\%$. Furthermore, we demonstrate the practical utility of our approach by showing that insights derived from skill slice analysis can generalize to held-out instances: when routing each instance to the model strongest on the relevant skills, we see a $3\%$ accuracy improvement over our 12 dataset corpus. Our skill-slices and framework open a new avenue in evaluation, leveraging skill-specific analyses to unlock a more granular and actionable understanding of model capabilities.

## 1 Introduction

Recent years have seen benchmarks evolve to keep up with ever-advancing models. While classical benchmarks tested specific capabilities, like recognizing digits (LeCun et al., 1998) or classifying sentiment (Bowman et al., 2015), modern benchmarks measure proficiency in numerous capabilities simultaneously, drawing questions of increasing difficulty from more diverse domains (Mialon et al., 2024; Yue et al., 2024; Wang et al., 2024). As the questions we test models on have grown more complex, aggregate performance measures provide less understanding about model proficiency in specific abilites. For example, as shown in Figure 1, we find that over a dozen benchmarks, GPT-4o, Gemini 1.5 Pro, and Claude 3.5 Sonnet (OpenAI, 2024; Gemini-Team, 2024; Anthropic, 2024) achieve overall accuracies within $0.4\%$ of one another, leaving an open question: Are these models all the same, or are valuable insights being averaged away?

Manual annotations of categories across instances (e.g. Ying et al. (2024) and Liu et al. (2023) include 'ability' tags) enable going beyond accuracy, but at a cost for benchmark creators that rises with the number and difficulty of test questions. This results in few (if any) and non-standardized annotations, restricting cross-benchmark aggregation, even though many benchmarks have large overlap in the (implicit) skills they test (Miao et al., 2020; Cobbe et al., 2021). Prior works show promise in automatically grouping inputs by attributes to extract image classification failure modes (e.g. 'fails

---

*Work done while interning at Microsoft Research

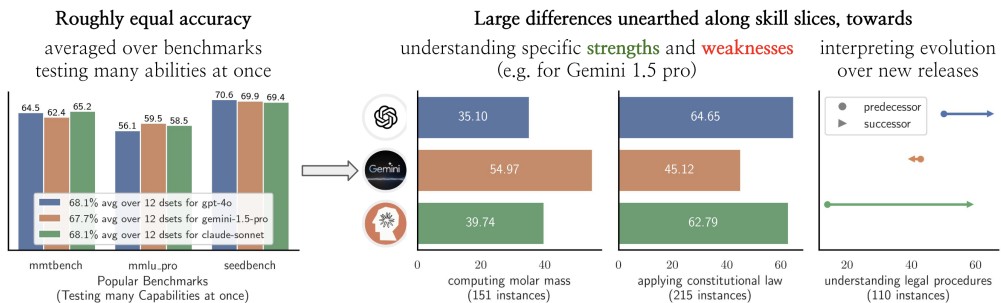

Figure 1: We leverage model-generated rationales to extract the skills relevant to any evaluation instance. Inspecting accuracy along *skill-slices* (instances drawn across benchmarks involving the same skill) surfaces fine-grained insights otherwise obfuscated by aggregate accuracy.

on *white* foxes') (Eyuboglu et al., 2022; Rezaei et al., 2024), but these methods fail to produce deeper insights on the underlying competencies that models use across contexts (e.g. 'fails on *counting*').

To this end, we propose a method to automatically discover *skills* related to any evaluation instance, toward a finer-grained understanding of model capabilities from existing benchmarks. We consider skills as latent features pertaining to "how" a model must operate – the steps involved in solving a given task – whereas a general attribute relates to "what" is being addressed – the observable characteristics of the instance. Moreover, the relevant skills for an instance may not be readily apparent through superficial examination, often only revealing themselves upon inspection of the solution. This complexity highlights the challenge of automatically inferring relevant skills and underscores the need to develop new approaches for their effective identification.

To this end, we propose to harness the *rationale*-generating ability of strong models to infer skills, as rationales (i.e. explained solutions) elucidate the steps involved in performing a task, and have had many benefits in the past (Wei et al., 2022; Kojima et al., 2022; Singh et al., 2024; Ehsan et al., 2019; Mitra et al., 2023; Hsieh et al., 2023). In our work, we use rationales not to directly analyze the rationale-generating model, but instead to annotate *data*, on which *any* model can be evaluated.

Specifically, given an evaluation instance, we instruct a strong model (e.g. GPT-4o) to generate a detailed step-by-step rationale, along with the skill applied in each step. We then parse skills per instance for numerous benchmarks and aggregate instances along ***skill-slices***, i.e. subsets of instances drawn from multiple benchmarks that all involve a specific skill, enabling measurement of model proficiency for that skill. After applying our technique to over $45k$ instances from a dozen popular modern benchmarks (primarily multimodal), we find *hundreds* of skills with slice sizes of $\geq 100$. To validate the relevance of our skills, we devise an automatic verification protocol and observe that our automatically extracted skills are relevant, even in cases where the annotating model fails in solving the query problem. We will release all our rationales, skill annotations, and skill-slices, which we term the *Skill-Index*, to the public at `github.com/microsoft/skill-slice-insights`.

Having verified the accuracy of our constructed skill-slices, we present multiple ways our skill annotations and skill-slices can be used to inform our understanding and usage of current large models. First, we conduct detailed skill-level analyses of 6 state-of-art models from the GPT, Gemini, and Claude families, **identifying practical insights into model strengths and weaknesses at previously underexplored granularities**. Despite roughly equal average accuracy across our corpus for GPT-4o, Gemini 1.5 Pro, and Claude 3.5 Sonnet, we discover skills for each model where accuracy far exceeds that of its counterparts, with gaps as large as $20\%$ (see Figure 1). For example, we find Gemini 1.5 Pro to be much stronger in math and science skills, like 'computing molar mass', while falling far behind for legal skills like 'applying constitutional law'. Next, we show that our **skill-slices allow for fine-grained understanding of improvements in new models within a model family**: while Gemini 1.5 Pro did not improve over v1.0 on 'understanding legal procedures', OpenAI and Claude models improved substantially, with Claude 3.5 Sonnet leaping nearly $50\%$ over Claude 3 Opus.

Further, our **skill-level understanding of models can be directly utilized for instance level model selection** based on skills required. When routing each query instance to a model with best performance on relevant skills, we see improvements in overall accuracy across a dozen benchmarks by

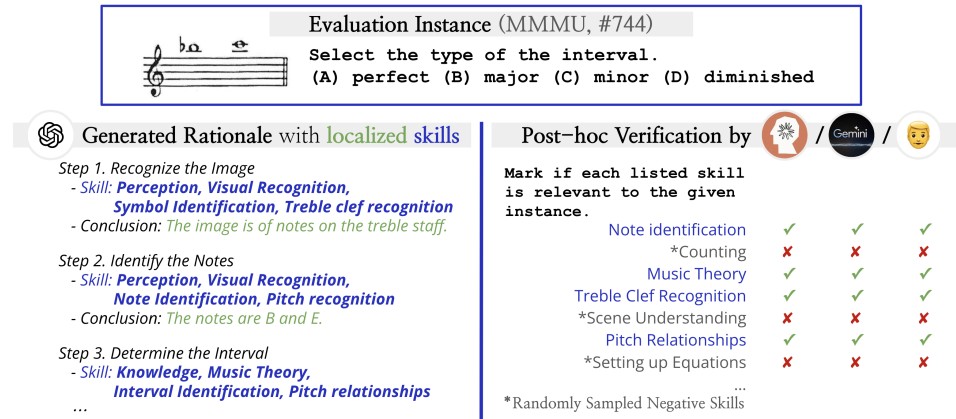

Figure 2: **(left)** Sample GPT-4o generated rationale: each skill is listed under multiple names of *cascaded granularity*, and localized to a specific step and concluding claim. **(right)** Annotated skills can be verified independently with a second model or human. We include randomly sampled negative skills and multiple verifiers to assure the quality of the verification, as detailed in 2.3.

3%, including gains of 3.5 to 7% on MMLU Pro (Wang et al., 2024). Finally, to assess a model's proficiency on a skill in isolation, we generate probing questions that directly target a specific skill. We find that model performance on the synthetic probing sets corroborates accuracies on our constructed skill-slices, further strengthening our findings.

Our skill based framework and findings advocate for a paradigm shift in foundation model evaluation, emphasizing the importance of skill-specific analysis to gain a more granular and actionable understanding of model capabilities. We summarize key contributions below:

- Automated Skill Inference: We present a scalable method that leverages model-generated rationales to recover (and validate) the underlying skills relevant to any evaluation instance.
- Novel findings from *Skill-Slices*: Analyzing skill-slices – sets of instances testing a common skill – reveals new fine-grained insights on the trade-offs across leading models.
- Skill Annotations for Popular Benchmarks: We release the *Skill-Index*, a dataset of instance-level skill annotations and rationales, along with code to easily expand to new benchmarks, providing a valuable resource for the research community.

## 2 CONSTRUCTING SKILL-SLICES USING MODEL-GENERATED RATIONALES

We now detail our method for inferring relevant skills for each evaluation instance and aggregating skill annotations across datasets to form *skill-slices* (i.e. a set of instances that share a relevant skill). We utilize *rationales*, or step-by-step solutions, generated by a strong model (GPT-4o) to facilitate the extraction of relevant skills. However, as evaluation instances are challenging by design and generating relevant skills requires a significant level of meta-reasoning, it is important to assess the quality of produced skills. Thus, we additionally present a multifaceted automatic approach to validating the relevance of extracted skills, which we confirm aligns well with human judgements.

### 2.1 SKILL EXTRACTION VIA RATIONALE PARSING

Given an evaluation instance, we prompt GPT-4o to generate a detailed rationale (i.e. step-by-step solution), where each step only involves the use of a single skill; figure 2 presents an example. We additionally instruct the model to, after each step, list the applied skill with *multiple* names of *cascading granularity*. For example, the skill 'treble clef recognition' is prepended with 'perception, visual recognition, symbol identification'[1]. Listing multiple names of different granularity per skill

---

[1] If some of these skills are unfamiliar to you, you're not alone! Many benchmarks today ask questions that are challenging to anyone without expertise in the relevant domain. This precisely motivates relying on stronger models to help us make sense of these benchmarks, as manual inspection has increasingly limited utility.

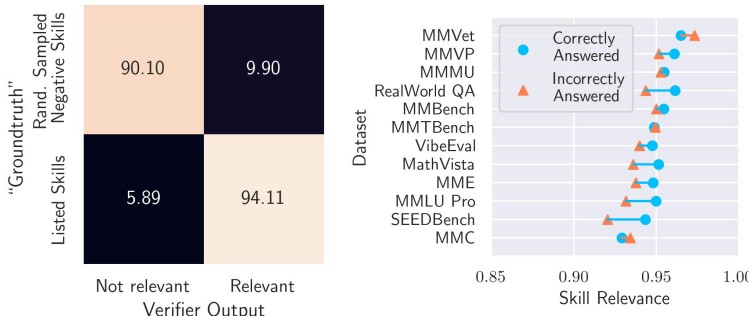

Figure 3: (**left**) Post-hoc verification shows GPT-4o-annotated skills are relevant, and that automatic verifiers are reliable, as they admit low rates of false positives (marking a randomly sampled negative skill as relevant). (**right**) Rate that GPT-4o-annotated skills are marked as relevant, separated by if GPT-4o correctly answered the underlying evaluation instance (blue) or not (orange). Empirically, annotated skills have high relevancy rates *even* when the annotator incorrectly answers the question.

is a simple way to increase the number of slices an evaluation instance falls into, leading to a greater quantity and diversity of skill-slices. Further, each granularity has unique and complementary advantages: fine-grained skills are more specific, while coarse-grained skills lead to slices with a larger number of samples, making an accuracy estimate over such slices more reliable. Also, in order to standardize the response structure, we include an in-context example in our prompt, which enables simple parsing of rationales to extract skills. See Appendix C for complete details.

Note that other methods could possibly be used to annotate relevant skills, like directly prompting GPT-4o to list them. We prioritized maximizing the total count and size of resultant skill-slices. In comparing the skills per instance obtained by direct prompting vs. our rationale parsing, we indeed find that rationale parsing results in a higher average count and diversity in grain (Appendix C.1).

## 2.2 CURATING SKILL-SLICES ACROSS BENCHMARKS

With rationale-parsing, we tag relevant skills for $46k$ evaluation instances from 12 benchmarks[2] that are popular (i.e. commonly featured in evaluations for recent model releases) and unsaturated (SOTA under $90\%$ accuracy), resulting in 690k total skills (128k unique) over 202k rationale steps. Importantly, we observe that **skills cut across benchmarks**. That is, many skill-slices of non-trivial size are formed when including instances across benchmarks, because *models re-use skills in diverse contexts*. Namely, 278 skill-slices have at least 100 unique instances each. This number rises to 332 after de-duplicating via a tight clustering[3] on the text embeddings of the skills. Aggregating across benchmarks enables us to analyze skills that are in the long tail for one benchmark, but get resurfaced when many different benchmarks are joined together. Indeed, slice count increases by $51\%$ when slicing across benchmarks instead of only drawing instances from one benchmark at a time.

## 2.3 AUTOMATED VALIDATION OF SKILL RELEVANCE

In the absence of fine-grained ground truth skill annotations that would confirm whether generated skills are relevant or not, we design an automatic validation approach. Namely, we propose two automated methods to directly validate the relevancy of listed skills: *post-hoc verification* and *inter-(skill)annotator agreement*. The latter checks the overlap in skills listed for the same instance by two different annotators. Here, we focus on the former and defer full details of both approaches to Appendix C.2. Post-hoc verification consists of a second 'verifier' model which inspects an evaluation instance with a list of skills and marks whether each skill is relevant or not. We form the list of skills by combining the skills for the instance with an equally sized list of *negative* skills, randomly sampled as follows: with $S$ denoting all skills from our corpus, negative skills for an

---

[2]See Appendix B for complete details on datasets annotated and the slice formation process.

[3]We employ a high minimum cosine similarity threshold of 0.95 per cluster to de-duplicate skills listed under slightly different names; e.g., 'trigonometry' and 'trigonometric calculation' have a similarity of 0.95.

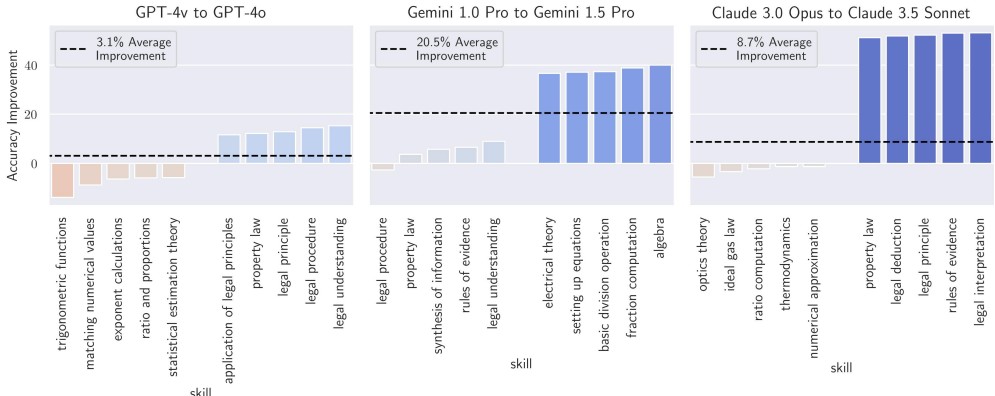

Figure 4: Skill-slices shed insight on how models evolve over new releases. For GPT and Claude models, skills related to law see the largest increases in accuracy, while Gemini models improve most in performing math and science skills.

instance $x$ with skills $S_x$ are drawn from $\{s \in S \mid \max_{s_i \in S_x} \text{sim}(s, s_i) \leq \tau\}$. Here, 'sim' is the similarity of two skills, computed via the cosine similarity of text embeddings from a frozen text embedder. The threshold $\tau$ is selected based on the text embedder so that no negative skill is equivalent up to paraphrasing to an annotated skill for the instance. We include negative skills as quality controls to ensure that the verifier model does not simply mark all skills as relevant.

Figure 3 shows the results of post-hoc verification of skills for 100 skills per dataset averaged over three verifier models (Claude 3.5 Sonnet, Gemini 1.5 Pro, GPT-4v[4]). We observe $94.1\%$ of skills annotated by GPT-4o are verified as relevant. A smaller human validation of $640$ total skills results in $95.7\%$ relevancy rate, as well as $92.8\%$ agreement with automatic verifiers, indicating that post-hoc verification is a reliable automated metric for assessing skill relevancy. The right panel shows that the relevancy rate is roughly equivalent, *regardless of if the annotator correctly answers the underlying evaluation instance*; crucially, this enables using automated skill annotations on instances that are interesting to evaluate on (i.e. hard enough to stump some models). Thus, while rationales may at times be partially incorrect[5], they can still be leveraged to shed insight on *data* at scale.

## 3 ANALYZING FOUNDATION MODELS WITH SKILL-SLICES

We now leverage the skill-slices obtained in the previous section to better understand and utilize frontier models from OpenAI, Google, and Anthropic. First, we identify skill-slices with vastly different accuracies across releases from the same family, as well as across families, which are obfuscated when inspecting overall accuracy. Then, having discovered models differ in their specific strengths and weaknesses, we show how overall accuracy can be improved by choosing the best model per instance or dataset, leveraging inferred skills and computed model-wise skill accuracies.

### 3.1 FINER-GRAINED INSIGHTS FROM EXISTING EVALUATION INSTANCES

**Understanding Model Evolution.** Figure 4 shows average improvement, along with skill-slices with greatest and least improvement, for the last two releases in the Open AI GPT, Google Gemini, and Anthropic Claude families[6]. For all model families, we find skills where improvement is more than twice the average, as well as skills where the model does not improve at all. Notably, both GPT and Claude models see greatest increases in skills related to law, with the Claude-Sonnnet 3.5 improving over Claude-Opus 3.0 by a staggering $\sim 50\%$ along numerous law skill-slices, suggesting improving legal abilities of their models may have been a recent priority for OpenAI and Anthropic.

---

[4]GPT-4v denotes GPT-4 Turbo 2024-04-09, and GPT-4o denotes GPT-4o 2024-05-13 throughout our paper.

[5]In fact, there's an error in the rationale in Figure 2: the notes are $B\flat$ & $C$. However, the skills are accurate.

[6]Because Gemini-1.0-pro is not multimodal, we present results for skill-slices curated from $12k$ language-only evaluation instances (instead of our entire corpus) with at least 100 instances within each slice.

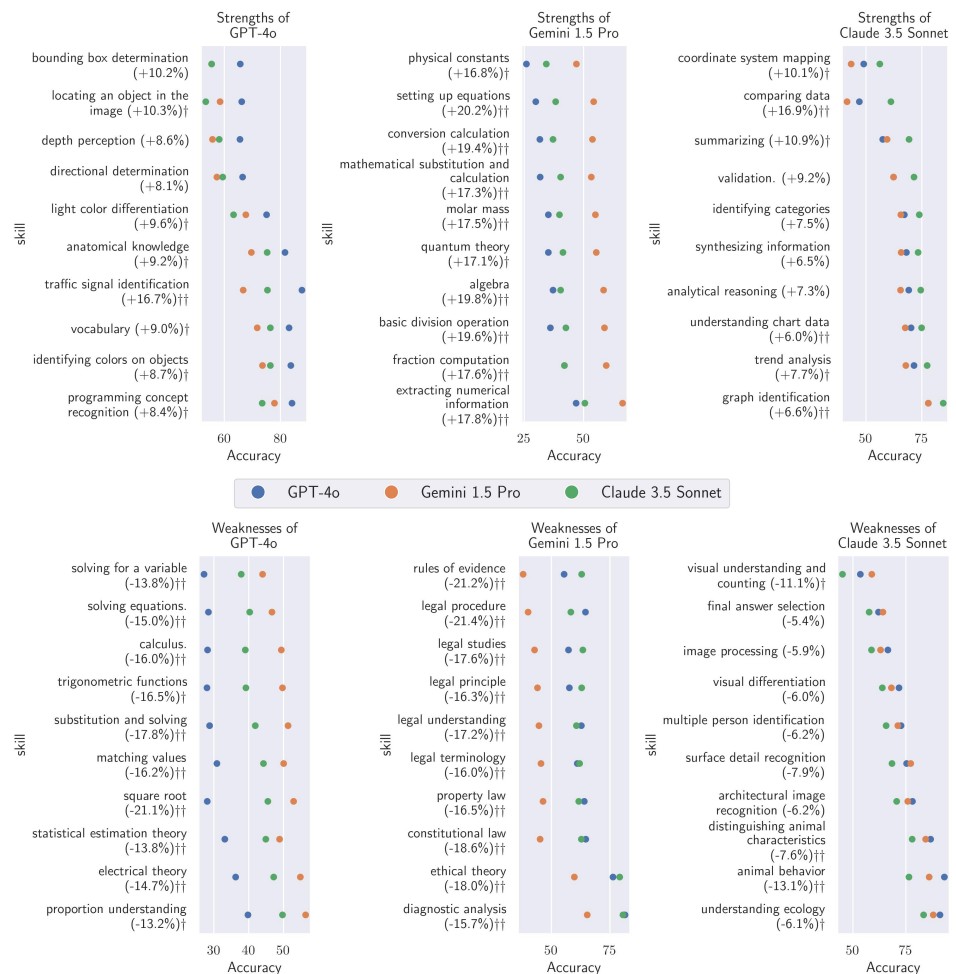

Figure 5: Unique strengths (**top**) and weaknesses (**bottom**) of GPT-4o, Gemini 1.5 Pro, and Claude 3.5 Sonnet, relative to one another. We present skills where each model's slice accuracy is highest / lowest (respectively) relative to the average of the other two model accuracies. † denotes significance ($p < 0.01$) via Wilcoxon signed-rank test; †† means both gaps vs. the other models are significant.

> Finding 1: **Skill-level improvements across model releases greatly differ (i) from average improvement and (ii) between families.** The biggest leaps for GPT and Claude families were for legal skills, while Gemini improved most for math and science skills (figure 4).

**Interpreting Trade-offs.** We now directly compare GPT-4o, Gemini 1.5 Pro, and Claude 3.5 Sonnet on all skill-slices with at least 100 unique evaluation instances. *While overall accuracies for the three models across our evaluation corpus falls within a half percent of one another, we observe differences as high as* 25% *for certain skills*, as shown in figure 5. Further, some patterns emerge: Gemini 1.5 Pro is distinctly strong for math and science, while GPT-4o's relative strengths pertain to visual skills like color differentiation and object localization, as well as the real-world skill of *traffic signal identification*, where GPT-4o is on average 16.7% more accurate. Interestingly, Gemini 1.5 Pro's greatest relative weaknesses nearly all concern law, which we found above to be the key skills that GPT-4o and Claude 3.5 Sonnet improved upon compared to their respective predecessors.

> Finding 2: **Models have unique strengths and weaknesses**: e.g., GPT-4o excels in visual skills, but lags behind on various math skills. Conversely, Gemini 1.5 Pro outperforms both other models on math and science skills, while falling far behind for legal skills (figure 5).

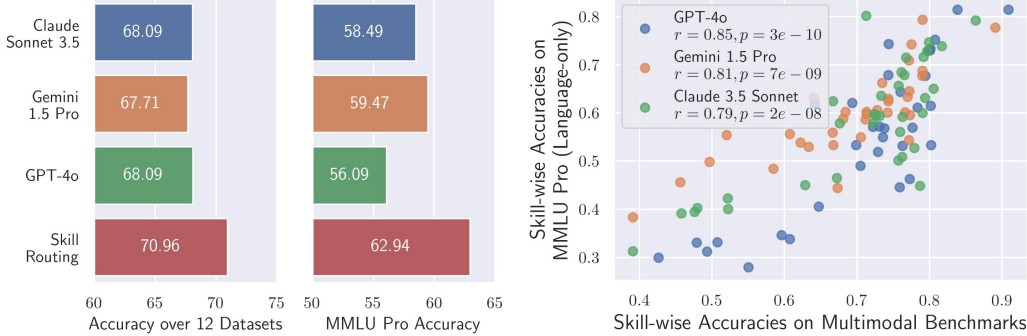

Figure 6: (**left**) A simple routing scheme that assigns each instance to the model with best accuracies for the skills relevant to that instance can lead to noticeable accuracy gains. (**right**) Accuracies over skill-slices can generalize to unseen data. In fact, skill-wise accuracies computed over multimodal benchmarks correlate strongly with accuracies obtained over a language-only benchmark.

Unfortunately, since we have limited visibility into the training data and processes for these private models, we cannot offer explanations for the strengths and weaknesses we observe. Nonetheless, this illuminates an advantage of our method: by annotating evaluation instances, **skill-slice analysis enables fine-grain insight on model capabilities, even when only granted black-box access**.

## 3.2 GENERALIZATION OF SKILL-SLICE INSIGHTS

Our skill-slice analysis operates under the premise that the insights drawn from inspecting sufficiently-large slices from a wide range of sources can *generalize* to new instances. If the insights generalize well, then understanding skill trade-offs between models can enable employing the models in a more calculated manner. That is, if we have knowledge of the skills relevant to a new instance, as well as each model's skill-wise strengths, we can improve accuracy by *routing* that instance to the model whose strengths are most aligned with those skills.

To test this, we route each instance in our corpus to one of GPT-4o, Gemini 1.5 Pro, and Claude 3.5 Sonnet, based on the skill annotations for that instance and the skill-wise accuracies per model computed over the remaining corpus (i.e. without the test instance). To obtain a single score per instance per model, we take a weighted average of skill-wise accuracies, where the weight for each skill is the inverse of its slice size (so to upweight finer-grained, more specific skills). As shown in the left panel of figure 6, routing increases accuracy by up to $3.2\%$ compared to each of the frontier models alone on 12 datasets combined, including improvements of 3.5 to $6.8\%$ for MMLU Pro (Wang et al., 2024).

We highlight the MMLU Pro results because the vast majority of the instances in the reference corpus (over which skill-wise accuracies are computed) are from multimodal benchmarks, while MMLU Pro is language-only, making it a good candidate to showcase the generalization of our approach. To study this deeper, we now partition each skill-slice based on if the instance comes from MMLU Pro or one of our multimodal benchmarks. Then, for each model we obtain two paired sets of slice-wise accuracies: one accuracy score per skill-slice, per partition. As shown in the right panel of figure 6, we observe strong correlations ($r \geq 0.79$, $p < 1e - 7$) between these two sets of scores for all three models. In addition to showing generalization, this result further validates our idea that skills pertain to a deeper property of a given instance (namely, what must be done to *solve* it) than surface level attributes, like its modality.

> Finding 3: **Skill-slice accuracies enable instance-wise model selection**. Routing each instance to the model strongest on the relevant skills results in a $3\%$ accuracy gain, indicating that skill-slice analyses offer generalizable insight (see figure 6).

We note that this proof of concept utilizes accuracies computed over slices defined by a single skill, not accounting for the added difficulty created when certain skills are used together. Nonetheless, our framework opens the door to studying these second order effects, e.g., by surfacing error-inducing slices defined by a *combination* of skills, as done by Rezaei et al. (2024) for image classifiers.

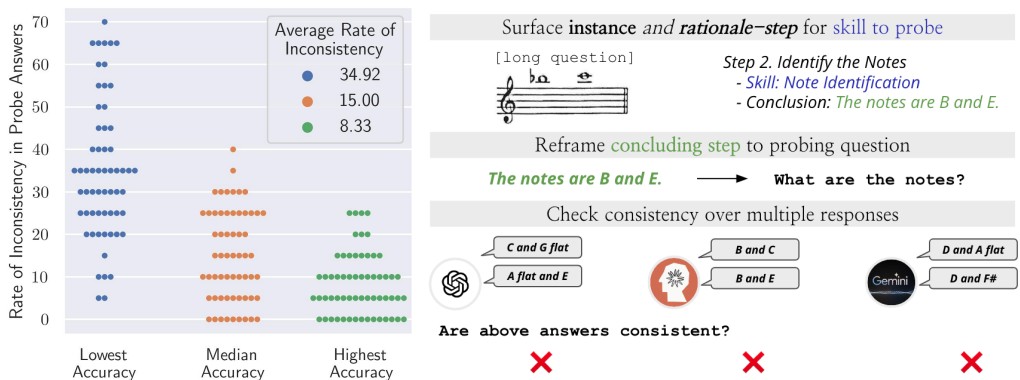

Figure 7: (**left**) Skills who's slices yield lowest model accuracy have the highest rates of inconsistency when probed. Each point corresponds to one model's consistency rate for a single skill. (**right**) Probe questions are generated by reframing rationale steps associated with a skill of interest.

## 4   CORROBORATING SLICE ACCURACIES WITH PROBING QUESTIONS

While skill-slices allow for approximating a model's proficiency at a skill by averaging accuracy over any instance where the skill is relevant, our rationale parsing framework enables a second independent analysis to more directly probe a *single* skill, without the effect of co-occurring skills.

Namely, rationale parsing *localizes* each skill to a specific step of the generated solution, as well as the resulting claim, which we instruct the rationale-generating model to include in its response. For example, as shown on the right in figure 7, the resultant claim to a step where the skill "[musical] note identification" is applied is "The notes are B and E". A claim can then be reframed as a question (e.g. "What are the notes?") probing a single skill, unlike the original question that often requires numerous skills and steps, any one of which could cause an incorrect answer. We use the rate of *inconsistency*, like in Wang et al. (2023), over multiple responses to each probing question as a measure of proficiency at the probed skill: if a model contradicts itself, it must have been wrong at least once.

We now probe three sets of 20 skills: those with lowest, median, and highest accuracy based on the analysis in section 3, to sample skills with a variety of slice accuracies. For each skill, we (i) generate 20 probing questions, (ii) obtain 5 responses per probe per model, and (iii) evaluate consistency of the responses with GPT-4o[7]. As shown in figure 7, models contradict themselves at a significantly higher rate for low accuracy skills, and **inconsistency rises as slice accuracy falls**, corroborating the skill-slice analysis. Overall, inconsistency correlates well with slice accuracy ($r = -0.675$; see Appendix E.2). Further, we find skills where all models contradict themselves more often than not, such as "[musical] note identification", "eye direction analysis", and "enumerating objects", the last of which is a well-documented limitation of VLMs (Yuksekgonul et al., 2023; Paiss et al., 2023).

> Finding 4: **Rationale-parsing enables targeting skills in isolation.** This independent analysis yields corroborating assessments of skill proficiency as skill-slice accuracies (figure 7).

In addition to independently verifying the insights from slice accuracies, probe inconsistency offers *complementary* signal: since some evaluation instances engage multiple skills, low accuracy over one skill-slice could be due to a deficiency on another highly co-occurring skill. However, if a model is proficient in a skill, it will always answer consistently (correctly) to probing questions. Thus, consistency on probing questions can be used to *refine* insights from skill-slice analyses, as a skill with low slice accuracy and high probe inconsistency is almost surely deficient. We discuss automatic diagnosis of skill deficiencies further in Appendix E.3, leaving it as a future application of our work.

---

[7]While we use GPT-4o, forming probing questions and evaluating consistency are likely simple enough to be done reliably by open-source LLMs (as both tasks are text-only). See appendix E.1 for all prompts used.

## 5 RETRIEVING EVALUATION INSTANCES FOR CUSTOM QUERY SKILLS

Finally, towards (a) direct comparison of our work to prior methods and (b) more flexible use of skill annotations, we introduce the task of *skill-based retrieval*: given an open-vocabulary query skill, can we retrieve relevant evaluation instances, so to build a custom skill-slice for specialized evaluation? This task boils down to defining an efficient metric that assigns a similarity score between a text query and an evaluation instance. Here, we focus on multimodal evaluation instances[8].

We consider **baselines** that leverage **a.** embeddings of the instance (either of just the image using CLIP Radford et al. (2021), just the text question, or the average of both image and text), **b.** attributes of the input, either inferred by GPT-4o[9] or provided as ground-truths by benchmark creators. In this latter case, we embed text attributes, along with the query skill, using a text encoder, with which we compute similarity of an attribute to the query skill as the cosine similarity of their embeddings. To obtain a single similarity score for the GPT-4o attributes baseline where multiple attributes exist per instance, we average the top 3 highest similarities[10]. Our method is identical to the attribute-based approach, except that we leverage annotated *skills* instead of attributes. The baselines encapsulate how prior methods propose to group inputs to go beyond overall accuracy (Eyuboglu et al., 2022; Rezaei et al., 2024), enabling direct comparison to our method.

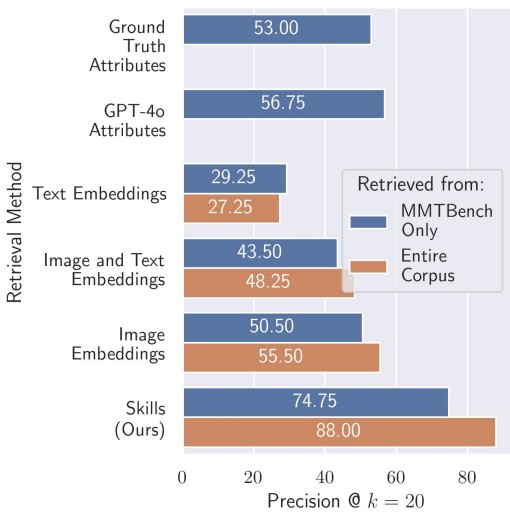

**Setup:** We take 20 skills from one benchmark (MMBench), and retrieve 20 instances per skill per method from two sources: **a.** a single other benchmark: MMTBench, selected because it has the highest number of ground-truth categories (162) out of our corpus, **b.** all benchmarks aside from MMBench, though we exclude ground-truth and GPT-4o inferred attributes baselines in the second case, for availability and cost reasons respectively. **Metric:** We evaluate each retrieved instance by presenting it along with the query skill to GPT-4o and prompting it to decide if the query skill is relevant or not; 'precision @ $k = 20$' is the fraction of the 20 retrieved instances deemed relevant.

Figure 8: Retrieving instances based on a query skill is far more effective when utilizing skill annotations than using direct representations of each instance, as done in prior work.

As shown in figure 8, retrieval using skill annotations far exceeds the precision for any other baseline in both settings, with improvements of 18-45.5% and 50.75-32.5% respectively. The attribute baselines are stronger than the embedding-based approaches, though obtaining attributes is also more expensive. The embedding-based baselines struggle, reflecting that frozen-embedding models may prioritize encoding surface-level information over skills, as skills may only become more apparent when inspecting the *solution* to the instance. We also find that searching over a broader corpus benefits the skill-based approach far more than either embedding-based approach. In summary, while allowing for direct searching over our corpus, these skill-based retrieval experiments underscore how our skill annotations offer novel signal, enabling the grouping of instances in ways that can be complementary to prior methods.

> Finding 5: **Skill annotations open the door to custom evaluation sets via skill-based retrieval.** Similarity of attributes or direct embeddings for two instances does not reflect the similarity of the underlying skills tested by each instance, leading to poor skill-based retrieval performance (figure 8).

---

[8]A multimodal instance consists of one **image** along with a **text** question.

[9]We obtain surface-level 'tags': the type of image, subject of the question, etc. See App. F.

[10]As in Moayeri et al. (2024a), we find top-3 averaging to be superior to top-1 or full averaging.

# 6   REVIEW OF LITERATURE

Prior efforts to gain **fine-grain insights** often involve intensive manual efforts to construct multi-task benchmarks (Liang et al., 2023; Wang et al., 2019; Srivastava et al., 2023; Al-Tahan et al., 2024; Balachandran et al., 2024) or datasets with annotations beyond ground-truth labels (White et al., 2024; Fu et al., 2023; Yu et al., 2024b; Lu et al., 2024). On top of quantifying specific abilities, extra annotations have illuminated biases (Buolamwini & Gebru, 2018; Moayeri et al., 2024b) and robustness issues (Idrissi et al., 2022; Koh et al., 2021). Automated alternatives include synthesizing benchmarks with known attributes (Zhang et al., 2024; Bordes et al., 2023) or automatically uncovering sub-groups within existing data (Luo et al., 2024; Murahari et al., 2023).The term *slice discovery* (or otherwise referred to as data *cohort* or *subgroup*) was coined (Chung et al., 2019; Eyuboglu et al., 2022) to describe these methods which enable comparisons (Eyuboglu et al., 2024) and error analysis (Rezaei et al., 2024; Slyman et al., 2023; Nushi et al., 2018; Singla et al., 2021; Dunlap et al., 2024). While these works focus on slices formed by surface-level attributes for visual recognition tasks, we curate *skill-slices*, which pertain to the underlying ability each slice requires from a model, and as such extend to more tasks and modalities.

**Skills** are akin to the 'ability' tags present in some modern benchmarks (Liu et al., 2023; Ying et al., 2024), though they can be finer (or coarser) grained (White, 1973) than what is typically annotated. The importance of acquisition of skills has been studied in human cognition (Gagne, 1962; Koedinger et al., 2023) as well as language models (Arora & Goyal, 2023). Yu et al. (2024a) utilizes skills as input to an LLM to generate challenging evaluation instances requiring a composition of the provided skills. Recent Murahari et al. (2023) and concurrent Didolkar et al. (2024) work also explore skill inference, albeit in more narrow domains, toward improved prompting or finetuning. This important shift of methods from attributes to skills is indeed motivated by the increasingly general-purpose nature of state-of-the art models. In this work, we scale up such analysis and insights by expanding it to 12 benchmarks and by introducing *rationale parsing* as a means to infer more skills.

**Rationales** have been studied extensively in the context of prompting (Wei et al., 2022; Kojima et al., 2022; Yao et al., 2023). Other works show the value of rationales as richer training signal (Hsieh et al., 2023; Mitra et al., 2023; Zelikman et al., 2022; Krishna et al., 2023) or model explanations (Ehsan et al., 2019; Hu & Yu, 2024; Huang et al., 2023), though some question their faithfulness to underlying model processes (Madsen et al., 2024; Fayyaz et al., 2024). We utilize rationales not to interpret a model directly, but instead to improve the relevance and diversity of the annotated skills (Singh et al., 2024), which become apparent in the solution steps present in rationales.

# 7   DISCUSSION AND FUTURE WORK

It is often said, "We can only improve what we can measure." As models grow in complexity and capability, traditional evaluation metrics like aggregate accuracy over entire datasets become insufficient. Simply averaging accuracy across datasets hides important insights needed to understand and enhance both the datasets and the models themselves.

To address this gap, our work introduced a skill-slice analysis approach for model evaluation. By examining model-generated rationales, we developed an automated method to extract the underlying skills required to solve individual evaluation instances. This shift from focusing on overall accuracy to conducting a more detailed analysis provides a more granular picture of a model's strengths and weaknesses. Furthermore, our reliance on a strong model to help humans understand evaluation data may prove to be increasingly pertinent as models get so strong that only a small handful of people will be qualified to provide manual annotations on evaluation instances. In this sense, our work relates to scalable oversight (Bowman et al., 2022), providing a new avenue to evaluation leveraging human-AI collaboration to understand models that will encapsulate advanced skills and knowledge.

Looking ahead, our approach can assist in creating more focused, skill-based evaluation datasets, by accelerating the process of discovering skills where models are collectively deficient or exhibit disparate performance. Our skill-based retrieval method can play a pivotal role in generating valuable training data for model development. By concentrating on specific skills, one can create more effective training sets that address identified weaknesses, thereby accelerating the advancement of AI models. This aligns with the principle that meaningful progress is achievable when we have clear insights, reinforcing the notion that we can only improve what we can measure.

## 8 ACKNOWLEDGEMENTS

We sincerely thank Siddarth Joshi, Alessandro Stolfo, and Natasha Butt for invaluable conversations and feedback throughout this project.

This project was supported in part by a grant from an NSF CAREER AWARD 1942230, ONR YIP award N00014-22-1-2271, ARO's Early Career Program Award 310902-00001, Army Grant No. W911NF2120076, the NSF award CCF2212458, NSF Award No. 2229885 (NSF Institute for Trustworthy AI in Law and Society, TRAILS), an Amazon Research Award and an award from Capital One.

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

# A    LIMITATIONS

A (somewhat surprising) key hypothesis our work hinges on is that models can articulate relevant skills, even for evaluation instances that they fail to answer correctly. While we empirically validate this claim for (a subset of) the skill annotations we present, it is possible that on harder benchmarks, the listed skills may no longer be accurate. Thus, we recommend always employing at least our post-hoc verification when collecting skill annotations on a new set.

Perhaps what is more likely is that while listed skills are relevant, they may be incomplete. Note that we do not present evidence for completeness in the main text. High overlap between two skill annotators is a promising sign (see Appendix C.2), the fact that overlap is not perfect suggests some skills may be missed. Moreover, most skills can always be broken down into smaller skills, making completeness perhaps an unnecessary objective.

Lastly, generating rationales is more expensive than directly prompting a strong model to list skills, or simply using a encoder to embed an entire instance at once. We show rationales have some advantages compared to each of these approaches in Appendix C.1 and Section 5 respectively, though the added cost of our method is worth mention.

# B    COMPOSITION OF SKILL-SLICES IN OUR STUDY

Here, we enumerate the datasets we compute skill annotations for, as well as provide more details on the resultant slices we form. All annotations and slices, which we call the *Skill-Index*, will be released to the broader community to enable skill-level analyses on models of their choice. We will also release code to easily add to the *Skill-Index*.

**Datasets**. We include datasets from 12 benchmarks in our study, consisting of 11 multimodal (image and text) datasets and 1 language-only dataset. These datasets are:

- MMLU Pro, a language-only benchmark by Wang et al. (2024), intended to be a harder version of MMLU (Hendrycks et al., 2021)
- MMMU, a multimodal benchmark with college level questions from many academic subjects intended to test 'expert' AI, by Yue et al. (2024)
- MathVista, a mathematics visual undersanding benchmark by Lu et al. (2024)
- MMC, a chart understanding multimodal benchmark by Liu et al. (2024)
- MMVP, a benchmark specifically focusing on failure modes of VLMs, by Liang et al. (2024)
- Many general multimodal benchmarks testing numerous abilities:
  - MMBench, by Liu et al. (2023)
  - MMTBench, by Ying et al. (2024)
  - MME, by Fu et al. (2023)
  - MMVet, by Yu et al. (2024b)
  - SEEDBench, by Li et al. (2023)
  - Realworld-QA, a benchmark that claims to test many realworld visual understanding questions, by xAI (2023)
  - VibeEval, by Padlewski et al. (2024) (also referred to as reka_vibe, as it was produced by the company Reka).

The two largest datasets in our benchmark suite are SEEDBench and MMLU Pro, which have 14k and 12k instances respectively. **For example instances from some of these benchmarks, along generated rationales, see Appendix H.**

We primarily sought out multimodal benchmarks, as many detailed benchmarks exist for language-only tasks. We aimed to select popular benchmarks by prioritizing benchmarks that were featured in recent reports for model releases or those tha had a large number of recent downloads on huggingface. We stress that benchmarks can easily be added to our corpus – our experiments suggest our prompt is effective in diverse settings.

**Forming skill-slices**. To consolidate the 128k unique total skills we extract, we perform a tight clustering over embeddings of the skill names. We use the `SFR-Embedding-2_R` model (Meng* et al., 2024), as it was the leader on the huggingface MTEB leaderboard at the time (Muennighoff et al., 2022). We utilize the fast clustering from `sentence-transformers` (Reimers & Gurevych, 2019) with a minimum similarity threshold of 0.95, intended to de-duplicate skills. One could use a lower threshold, though in that case, naming a resultant skill cluster would be more challenging – we simply take the skill name in the cluster that it closest to the centroid, though we qualitatively observe nearly all skills in the cluster have very similar names. To be sure we have completely de-duplicated, we run the clustering algorithm until all but at most 0.5% of cluster names have a similarity below 0.95.

Throughout our paper, we use skill-slices with at least 100 instances, though this number is arbitrary. Picking a lower threshold increases slice count (often including finer-grained slices), but accuracy over theses slices may be less reliable.

## C DETAILS ON SKILL EXTRACTION AND VALIDATION

We present the prompt used to generate rationales below. It features detailed instructions and an in-context example. We generate this prompt by iteratively asking ChatGPT to refine the prompt based on specific desiderata, along with manual tweaks along the way. The in context example is also generated by GPT-4o; it was produced using an earlier version of the prompt, and then modified slightly to correct some errors. Note that by GPT-4o, we always mean GPT-4o 2024-05-13.

---

**Prompt to generate rationales (part 1)**

**System Prompt: Detailed Step-by-Step Response with Skills Labeled**
**Objective:** List the skills and evidence for each step to solve the given question. Do so by responding to each question in a detailed, step-by-step manner, where each step utilizes only a single skill. *Clearly state the skill being used and the evidence applied at each step.*
**Guidelines:**

1. **Single Skill Per Step:**

   - Ensure each step involves only one skill. Skills can be categorized with multiple names, each getting more specific. For example:
     - **Knowledge:**
       * Music Theory, Scale Identification, Reciting F Major scale
       * Biology, Photosynthesis Process, Calvin cycle
       * History, World War II Events, Battle of Stalingrad
     - **Perception:**
       * Visual Recognition, Musical Note Identification
       * Chart understanding, information extraction, determining the length of a bar in a bar chart
       * Visual understanding, Spatial reasoning, Understanding depths of objects
       * Auditory Recognition, Sound Identification, Recognizing a child's voice
       * Video Analysis, Episodic memory, Remembering where an object was placed
     - **Reasoning:**
       * Logical Deduction, Process of Elimination
       * Mathematical Calculation, Algebraic Manipulation, Resolving fractions

2. **State the Skill:** Clearly identify and state the skill being used in each step.

3. **List all pieces of relevant evidence:**

   - Specify the evidence used in each step. This could include:
     - Information from the question (e.g., a specific phrase).
     - Data or findings from previous steps. *List these findings clearly if they are required context for the claim of the step!*
     - Regions in an attached image, if applicable, *clearly stating WHERE in the image the information is being extracted from.*

4. **State the Conclusion of the Step.**

5. **List your final answer after your step-by-step explanation in a new line formatted as 'ANSWER: {}'**

Here is an example of the desired response structure given a question.

---



**Prompt to generate rationales (part 2)**

**Example Question:** "Based on the following image, can you determine what the two sounds belong to? \image 1
A. #F major scale
B. c melodic minor scale
C. b harmonic minor scale
D. E melodic major scale"
**Example Response Structure:**

1. **Step 1: Recognize the Type of Image**

   - **Skill:** Perception: Visual Recognition, Image Classification, Sheet music recognition
   - **Evidence:** The image has lines and symbols, which resemble sheet music.
   - **Conclusion:** The image is of sheet music.

2. **Step 2: Identify the Clef**

   - **Skill:** Perception: Visual Recognition, Symbol Identification, Treble clef recognition
   - **Evidence:** There is a staff in the middle of the image, and on the left, there is a treble clef.
   - **Conclusion:** The music shown is on the treble clef.

3. **Step 3: Identify Symbol Positions from the Image**

   - **Skill:** Perception: Visual Recognition, Spatial reasoning, Locating musical symbols
   - **Evidence:** Two notes are present in the middle of the image: one with a sharp in the second space from the bottom, and one in the space above the top line.
   - **Conclusion:** The notes are F# and A.

4. **Step 4: Read Provided Options**

   - **Skill:** Knowledge: Reading Comprehension, Understanding the question
   - **Evidence:** The provided options are written after the question.
   - **Conclusion:** The options are F# major scale, c melodic minor scale, b harmonic minor scale, and E melodic major scale.

5. **Step 5: Eliminate Incorrect Options**

   - **Skill:** Reasoning: Logical Deduction, Process of Elimination
   - **Evidence:**
     - The notes shown are F# and A, so they must be in the correct scale.
     - F# major includes F# and A.
     - c melodic minor does not include F#.
     - b harmonic minor does not include F#.
     - E melodic major is not a standard scale.
   - **Conclusion:** c melodic minor, b harmonic minor, and E melodic minor are incorrect.

6. **Step 6: Conclude the Correct Scale**

   - **Skill:** Reasoning: Logical Deduction, Conclusion
   - **Evidence:** F# major scale is the correct answer, as it includes both F# and A.

   **ANSWER: A**

**Final Note:** Ensure that the skills are described in detail: **LIST AT LEAST THREE NAMES FOR EACH SKILL.**



## C.1  ABLATIONS ON RATIONALE PARSING FOR EXTRACTING SKILLS

We now perform an ablation study to compare how skills inferred by rationale parsing (our proposed method) differ from those obtained by directly prompting a strong model to list relevant skills. Specifically, the 'direct prompting' alternate method consists of presenting an evaluation instance along with the prompt "List skills that are relevant to the given question. Do not answer the question, only provide the list of skills. Be detailed, but only use short specific phrases throughout your response – only use a few words per list item. I will provide you with an example first." As the prompt suggests, we additionally provide an in context example, primarily to ensure standard response format to facilitate automatic parsing of the skill annotator model's outputs. We compare the annotated skills produced by GPT-4o over 1195 instances from all 12 benchmarks (roughly 100 per benchmark) using direct prompting and rationale parsing.

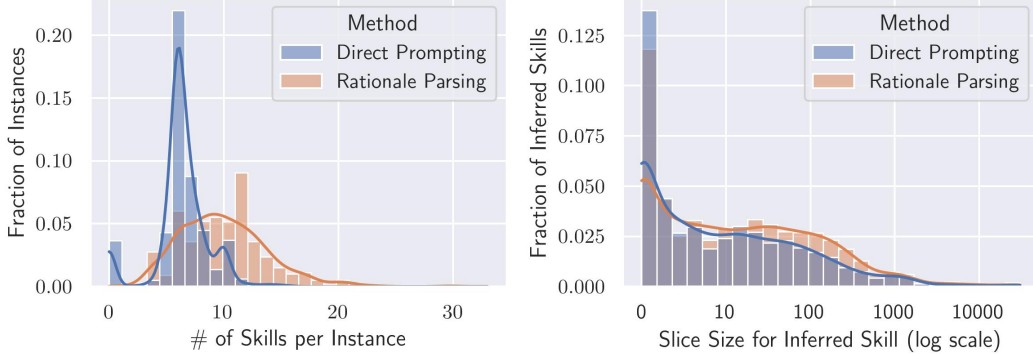

Figure 9: (**left**) Rationale parsing results in a significantly higher number of annotated skills per instance than using direct prompting. (**right**) Skills inferred with direct prompting are generally finer-grained, resulting in a higher fraction of very small slices (e.g. with < 10 instances), and a lower fraction of all other slice sizes (see table 1 for exact numbers).

| Slice Size
Skill Inference Method | $[0, 10)$ | $[10, 25)$ | $[25, 50)$ | $[50, 100)$ | $[100, 250)$ | $[250, 1000)$ | $\geq 1000$ |
|---|---|---|---|---|---|---|---|
| Direct Prompting | 2494 | 525 | 341 | 318 | 288 | 185 | 123 |
| Rationale Parsing | 2399 | 560 | 451 | 409 | 491 | 290 | 164 |

Table 1: Comparison of slices formed by skills inferred via direct prompting vs. rationale parsing. Direct prompting results in a higher number of skills that are very specific to the question, resulting in more slices with less than 10 instances than rationale parsing. Aside from this hyper-fine grain, rationale parsing results in more slices at every level of granularity, with largest improvements in medium grain slices. Details in section C.1.

First, we observe that **rationale parsing results in a substantially higher number of unique skills annotated per instance than direct prompting**. The left panel of figure 9 shows the distributions of number of annotated skills per instance with the prompting methods. On average, we obtain 9.98 skills per instance with rationale parsing, a 58% relative improvement compared to the average of 6.32 skills per instance with direct prompting. We note that for about 4% of instances, direct prompting leads to 0 inferred skills because the skill annotator directly answers the instance instead of listing the relevant skills. In the rationale parsing method, the skill annotator both answers the given instance and list skills simultaneously, alleviating issues with conflicting instructions.

Next, we seek to compare the granularity of inferred skills from the two methods. Recall that there is no single 'optimal' granularity: finer-grained skills are more descriptive, while coarse-grained skills result in larger skill-slices, which in turn improve the reliability of an average accuracy taken across such a slice. Thus, ideally, we can obtain skill annotations of various granularities. However, skills that are exceedingly fine-grained cannot be used, as skill-slices consisting of only a few instances may not have sufficient sample size to offer reliable measurements of model proficiency.

To proxy a skill's granularity, we count the number of instances in our corpus annotated (via rationale-parsing) with a highly similar skill (i.e. similarity of at least 0.95 between text embeddings of the skill names). Figure 9 (right) and table 1 show the distributions of the # of relevant instances per skill inferred via direct prompting and rationale parsing. First, we observe a larger fraction of skills inferred with direct prompting to be relevant for a very small number of instances, suggesting that **many of the skills inferred by direct prompting may be too fine-grained to result in sufficiently large skill-slices**. At other levels of granularity, rationale parsing consistently yields more skills for each grain. For example, 491 skills inferred with rationale parsing have between 100 and 250 relevant instances, while only 288 skills inferred with direct prompting have the same number of relevant instances – a reduction of 41%. We note that the improved diversity of grain of inferred skills using rationale parsing vs. direct prompting may in part be attributed to our instruction to list skills with multiple names of cascading granularity.

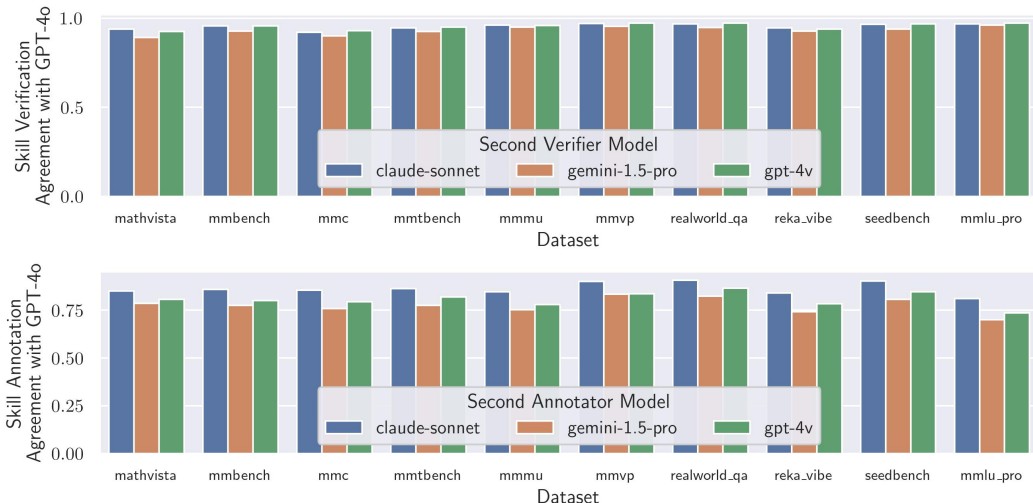

Figure 10: Diverse models agree when verifying the relevancy of an annotated skill (**top**) and annotating skills given an instance (**bottom**).

In summary, our proposed rationale parsing method results in more annotated skills per instance and greater diversity in granularity of annotated skills, when compared to direct prompting. Also, we qualitatively observe both methods to yield skills that are overwhelmingly relevant to the given instance – we do not quantitatively test this facet, as we do not expect much difference in the relevance of skills annotated by either method. Another aspect worth noting is that rationale parsing enables skill localization, which offers a second, complementary avenue to corroborate and refine our skill slice findings (see sections 4 and E.3).

We emphasize that better methods to extract skills certainly may exist. Here, we present one method and demonstrate the utilities it may hold. Many of these utilities can be extended and potentially expanded by improving our skill extraction method. Having demonstrated that skill inference is possible and useful, we leave further iteration on skill inference and its applications to future work.

Lastly, we again note that rationales are leveraged not to directly gain insight on the rationale-generating model, but instead to annotate evaluation data, with which any model's proficiencies can be studied more closely. This distinction is important as it side-steps the potential concern of rationale faithfulness. While the rationale-generating model may not be faithful in its written chain of thought, we are solely concerned with whether or not the skills listed in this chain of thought are relevant to the underlying evaluation instance. Thus, our validation focuses on understanding relevance of skill annotations to data, and not to hidden model internal mechanisms.

## C.2    DETAILS ON AUTOMATIC VALIDATION

We now detail our second automatic validation approach: inter-(skill)annotator agreement. As the name suggests, we simply extract skills by feeding the same prompt to a new rationale-generating model. Namely, we try Claude 3.5 Sonnet, Gemini 1.5 Pro, and GPT-4 Turbo 2024-04-09. We compute agreement between two sets of skills listed by different models as the overall fraction of skills such that a match exists in the other set of skills. Here, we define two skills being match if their text embedddings have a cosine similarity of at least $0.85$. Note that this is also the threshold we set for $\tau$ in sampling 'negative' skills for post-hoc verification. This threshold is lower than our clustering threshold because in this case, we do not want to admit paraphrased versions of the same skill, where as in our clustering, we only wish to ensure extremely high similarity within each slice, since it ultimately only takes on one name. Figure 10 shows the inter-annotator agreement, both across different skill annotators (bottom) and verifiers. For verifiers, agreement is very high ($> 0.9$) across the board, while agreement is slightly lower for skill annotators. We believe this may relate to the fact that our skill discovery procedure may not result in a complete list of skills, which we note in section A. We argue precision (skill relevance) is more important than recall (completeness), as

| Dataset (# of instances) Model | SEEDBench (14232) | MMLU Pro (12032) | MMBench (4329) | MMTBench (3123) | MME (2374) | MMC (2126) | MathVista (1000) | MMMU (900) | RealWorld-QA (765) | MMVP (292) | VibeEval (269) | MMVet (218) |
|---|---|---|---|---|---|---|---|---|---|---|---|---|
| Gemini 1.5 Pro | 69.94 | 59.47 | 84.18 | 62.36 | 78.64 | 75.40 | 56.10 | 56.33 | 68.10 | 74.00 | 36.43 | 60.55 |
| Claude 3.5 Sonnet | 69.43 | 58.49 | 84.50 | 65.17 | 84.33 | 82.27 | 48.60 | 58.67 | 71.76 | 65.00 | 44.98 | 59.63 |
| GPT-4o | 70.62 | 56.09 | 87.46 | 64.52 | 82.73 | 77.66 | 50.70 | 60.44 | 75.95 | 85.62 | 49.81 | 62.84 |
| Random choice | 69.99 | 58.02 | 85.38 | 64.03 | 81.90 | 78.44 | 51.80 | 58.48 | 71.94 | 74.78 | 43.74 | 61.01 |
| Skill Routing (ours) | 71.81 | 62.94 | 87.13 | 65.42 | 85.89 | 82.13 | 55.50 | 58.89 | 76.34 | 80.82 | 49.07 | 59.63 |

Table 2: Per-dataset accuracies for each of the three foundation models we study, as well as accuracies achieved when randomly selecting a model per instance or when routing each instance to the model strongest on the relevant skills.

omitted skills do not affect the veracity of skill-slice accuracy (so long as they are omitted randomly), while incorrect skills do corrupt the signal.

# D    DETAILS ON ROUTING PROOF OF CONCEPT

We route each instance independently, based on skill-slice accuracies computed over the remaining corpus (i.e. everything except for the current instance). For an instance, we first take all of its listed skills, and then remove any for which we do not have a skill slice with at least 100 instances on the remaining corpus. If no skills remain, we default to Claude 3.5 Sonnet, which is (barely) the best model over the entire corpus. For the remaining skills, we consolidate accuracies over those slices to obtain a single score per model via a weighted average. Namely, the weight for each skill is the inverse of the number of instances in that skill slice. This way, more specific (i.e. lower frequency) skills for the instance carry higher weight.

This routing protocol is very simple and does not account for numerous sources of variance, such as different levels of difficulty across instances for the same skill, and how skills can be harder when applied in the context of other skills. Nonetheless, we see accuracy improvement using our simple scheme, suggesting that skill-slice insights generalize.

To provide a full breakdown of the effectiveness of our simple approach, we present the per-dataset results in table 2. We show accuracy per dataset for each of the three foundation models in our study. Also, we show the accuracy obtained with skill routing (our method) or by selecting a model at random – in the latter case, we show the expected accuracy. For 11 out of 12 datasets, skill routing yields higher accuracy than random model selection, with the sole exception occurring for the smallest dataset in our study (where an accuracy measure may not be as reliable due to the smaller sample size). The accuracy gains exceed $3\%$ for 7 of the 12 benchmarks studied. Compared to the best model over our entire suite (Claude 3.5 Sonnet), skill routing improves accuracy for 10 out of 12 benchmarks, with a tie in one benchmark. GPT-4o, however, beats skill-routing for 5 benchmarks, though three of these are the smallest benchmarks in our study.

Overall, we find that our simple approach to skill routing offers accuracy gains under many settings. We interpret this as positive signal in support of the generalizability of skill-slice insights, as well as a direct potential application of our analysis to improve performance. As mentioned above, the simplicity of our scheme may limit its effectiveness. Potential avenues to improve this approach include considering accuracy over slices formed by *multiple* skills used together, or perhaps by training a lightweight router to predict the best model given skill annotations (akin to an interpretable mixture of experts). We leave these explorations to future work.

# E    DETAILS ON PROBE CONSISTENCY ANALYSIS

We now provide complete details for our probe consistency analysis.

## E.1    GENERATING PROBING QUESTIONS AND CHECKING CONSISTENCY

To generate probing questions and check consistency, we require multiple calls to an LLM – we use GPT-4o. Importantly, neither task requires a multimodal LLM, thanks to skill localization.

First, we rephrase a claim linked to a skill of interest with the below prompt. Notice that we form probing questions in a few slightly varied way, including two yes or no questions testing contradictory questions. Qualitatively, this increases the chance that a model contradicts itself when it is unsure, as opposed to giving the same exact incorrect answer (which our consistency analysis cannot detect). We omit prompts for answering probe questions and checking consistency, as these are straightforward and can be viewed in our code release.

---

**Probing question generation**

I will provide a specific skill and an excerpt from a step-by-step solution to some problem. Isolate the central claim that best pinpoints the given skill (next to *SKILL TO PINPOINT*). Then, state that claim and generate three questions that can be used to verify the claim, using the following strategy:

- Q1 should repeat the claim as a yes or no question.

- Q2 should create a close but contradictory claim, and ask that as a yes or no question. *Do not use negations*.

- Q3 should be open-ended, whose answer is contained in the claim alone.

ANSWER IN THE FORM OF A PYTHON DICTIONARY, as shown in the following example.

**Example Response Structure**:
*SKILL TO PINPOINT*: identifying the longest bar
*EXCERPT*: **Step 1: Identify the Top Companies from the Image**

- **Skill:** Perception: Visual Recognition, Bar chart analysis, Identifying the longest bars

- **Evidence:** In the chart, the two companies with the longest bars are AG Insurance and AXA.

- **Conclusion:** AG Insurance and AXA are the companies with the highest market shares.

**CLAIM AND PROBE QUESTIONS**:
```
{
"CLAIM":  "The two companies with the longest bars in the
chart are AG Insurance and AXA.",
"Q1":  "Are the two companies with the longest bars in the
chart AG Insurance and AXA?",
"Q2":  "Are the two companies with the longest bars in the
chart Allianz and AG Insurance?",
"Q3":  "Which companies have the longest bars in the chart?"
}
```

---

## E.2 Correlation of Probe Inconsistency and Skill-slice Accuracy

In figure 11, we plot probe inconsistency rate vs slice accuracy directly and for each model separately, as opposed to the consolidated violin plot we show in figure 7. We observe reasonably high correlations for all three models, especially given that probe inconsistency rate can only be a factor of $0.05$, as our implementation checks 20 claims per probed skill.

## E.3 Combining Slice and Probing Analyses Towards Automatic Diagnosis of Skill Deficiencies

Lastly, we briefly discuss an end-to-end pipeline for automatically diagnosing skill deficiencies. As shown in figure 12, a two-stage pipeline can be engineered that first flags skills with low slice accuracy, and then probes those skills. These two stages are complementary in two ways: first, they admit different kinds of errors. If we interpret a deficient skill as a 'positive', low skill-slice accuracy can admit false positives due to the presence of a highly co-occurring deficient skill. However, a deficient skill will necessarily have low skill-slice accuracy. Probe inconsistency can help remove these false positives, as a model will always answer consistently (i.e. correctly) to probe questions if the model is proficient at a skill.

The order of these components is also motivated by the fact that probing requires many more LLM calls, and as such is more expensive than skill-slice analysis. Nonetheless, if cost is not a factor, this procedure could potentially uncover skill-level deficiencies at scale.

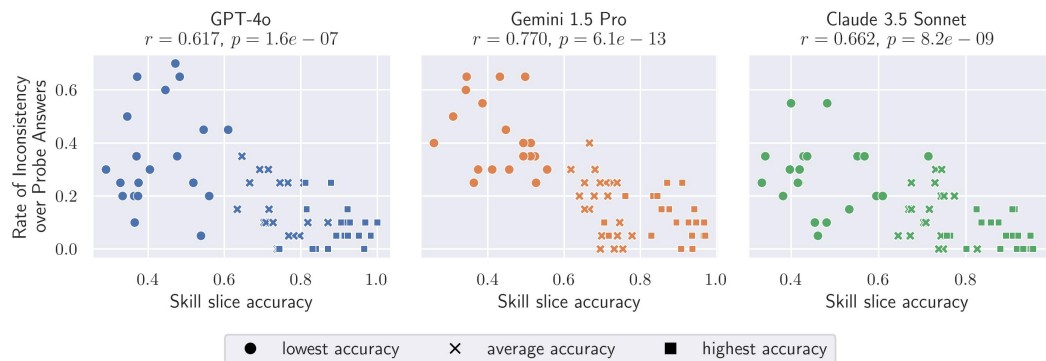

Figure 11: Rate of inconsistency over probe answers vs. slice accuracy per model per skill, as portrayed as a violin plot in Figure 10. For all models, inconsistency negatively correlated with slice accuracy, with an overall Pearson's $r = -0.675$.

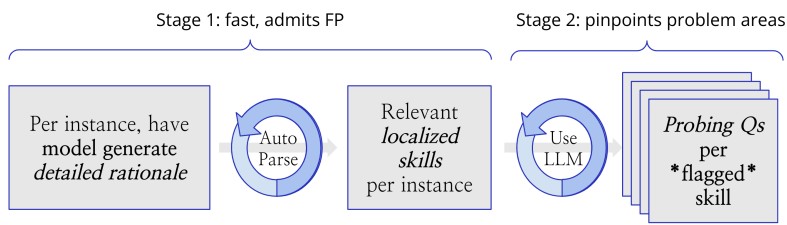

Figure 12: Potential framework for automatic diagnosis of skill deficiencies, leveraging the complementary nature of skill-slice analysis and probe inconsistency.

## F   DETAILS ON SKILL-BASED RETRIEVAL

We use the same text encoder (`SFR-Embedding-2_R`) for any text features, including embedding skills and attributes. We use CLIP ViT-L/14 Radford et al. (2021) for image features. To generate attributes, we prompt GPT-4o to "List attributes describing the content of given question. **Do not list skills needed to solve the question**, just attributes of the content". We additionally instruct GPT-4o to avoid answering the question we wish for it to attribute.

## G   PROMPT USED TO GENERATE MODEL OUTPUTS

We note that we employ a simple prompt in obtaining model outputs on the benchmarks we analyze. Namely, we ask models to "Be concise. Write 'ANSWER: ' followed by your answer. If multiple choices are given, only provide the correct letter." As is well documented, accuracies can be improved by engineering stronger prompts, including with model-specific strategies. We opt to choose a single standard prompt for all models, so to avoid giving any single model an advantage due to unequal amounts of time spent on prompt optimization. We leave investigation of these for future work and study models in a simple setting for now.

## H   QUALITATIVE EXAMPLES

We now present numerous qualitative examples so that readers may more easily comprehend the generated rationales and resultant skill annotations that form the basis of our method. Namely, we show one example instance for each benchmark from a subset of our evaluation corpus, along with the corresponding generated rationale. For cosmetic reasons, we choose the instances with the shortest question length. Note that even for these shorter questions, rationales can be long. The skill annotations are written as the first bullet under the rationale step they pertain (or are *localized*) to, with each skill articulated with multiple comma-separated names of cascading granularity.

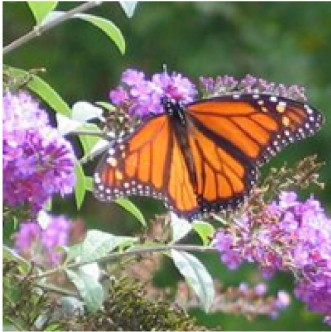

Are the butterfly's wings closer to being open or closed? (a) Open (b) Closed

### Example rationale for MMVP instance

**Step 1: Identify the Duck in the Image**

- **Skill:** Perception: Visual Recognition, Animal Identification, Identifying ducks
- **Evidence:** The image shows a bird with recognizable features of a duck including a green head and a distinct body pattern.
- **Conclusion:** The bird in the image is indeed a duck.

**Step 2: Locate the Duck's Beak in the Image**

- **Skill:** Perception: Visual Recognition, Feature Identification, Locating beak
- **Evidence:** The front part of the duck's head, where the beak should be, is clearly visible in the bottom right of the image.
- **Conclusion:** The duck's beak is located.

**Step 3: Determine Visibility of the Entire Beak**

- **Skill:** Perception: Visual Assessment, Fullness Identification, Checking visibility of parts
- **Evidence:** The entire shape and contour of the duck's beak can be seen without any part hidden or cut off by the edge of the image or by any obstruction.
- **Conclusion:** The entire beak of the duck is visible in the picture.

**ANSWER:** (a) Yes

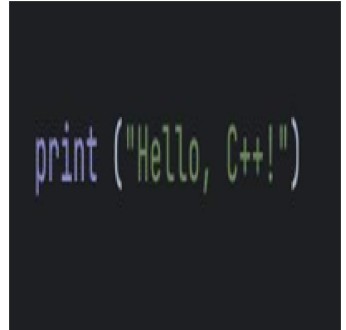

Is a python code shown in the picture?
A. Yes
B. No

### Example rationale for MMC instance

**Step 1: Recognize the type of content in the image**

- **Skill:** Perception: Visual Recognition, Text Identification, Code Snippet Recognition
- **Evidence:** The image contains text that appears to be a line of code.
- **Conclusion:** The content in the image is a snippet of code.

**Step 2: Identify the programming language structure**

- **Skill:** Knowledge: Programming, Syntax Recognition, Python Syntax
- **Evidence:** The code 'print("Hello, C++!")' uses the 'print' function with parentheses and a string in quotes, which matches the syntax of Python.
- **Conclusion:** The code structure matches Python syntax.

**Step 3: Evaluate the print statement in Python**

- **Skill:** Knowledge: Programming, Python Functions, Print Statement
- **Evidence:** In Python, 'print' statements are used to display text or variables, and the code snippet does exactly that with 'print("Hello, C++!")'.
- **Conclusion:** The code is a valid Python 'print' statement.

ANSWER: A

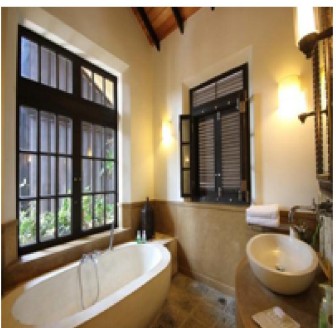

How many towels are in the image?
A. One
B. Two
C. Three
D. Four

### Example rationale for SEEDBench instance

**Step 1: Recognize and Identify Relevant Objects**

- **Skill:** Perception: Visual Recognition, Object Identification, Towel recognition
- **Evidence:** The image shows various objects within a bathroom setting, including a sink, a bathtub, and items placed nearby.
- **Conclusion:** Towels must be identified within the bathroom scene.

**Step 2: Locate Towels in the Image**

- **Skill:** Perception: Visual Recognition, Spatial Reasoning, Identifying multiple objects
- **Evidence:**

  - On the right side of the image, near the sink, there are folded items that look like towels.
  - On the corner of the bathtub's structure, to the left of the sink, there's a stack of what appears to be towels.

- **Conclusion:** Towels are located near the sink and on the corner of the bathtub structure.

**Step 3: Count the Towels**

- **Skill:** Reasoning: Quantitative Analysis, Counting Objects
- **Evidence:** By the sink, there appears to be a single stack composed of three folded towels.
- **Conclusion:** There are a total of three towels visible in the image.

## Example rationale for RealWorld-QA instance

**Step 1: Recognize the Type of Objects**

- **Skill:** Perception: Visual Recognition, Object Identification, Electronic device recognition
- **Evidence:** The objects are electronic devices.
- **Conclusion:** The three objects are electronic devices placed side by side.

**Step 2: Compare Object Heights**

- **Skill:** Perception: Visual Assessment, Height Comparison
- **Evidence:** The object in the middle appears to be shorter in height compared to the objects on the left and right.
- **Conclusion:** The object in the middle is shorter in height.

**Step 3: Compare Object Widths**

- **Skill:** Perception: Visual Assessment, Width Comparison
- **Evidence:** The object in the middle appears to be narrower in width compared to the objects on the left and right.
- **Conclusion:** The object in the middle is narrower in width.

**Step 4: Compare Object Depths**

- **Skill:** Perception: Visual Assessment, Depth Comparison
- **Evidence:** The object in the middle appears to be thinner in depth compared to the objects on the left and right.
- **Conclusion:** The object in the middle is thinner in depth.

**Step 5: Overall Size Comparison**

- **Skill:** Reasoning: Logical Deduction, Aggregation of Dimensions
- **Evidence:** The object in the middle is shorter in height, narrower in width, and thinner in depth compared to the other two objects.
- **Conclusion:** The object in the middle is the smallest overall.

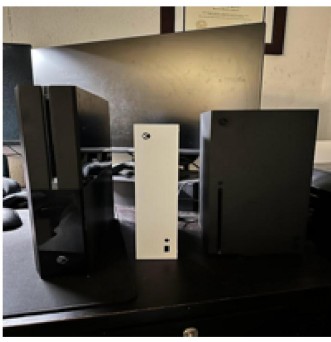

Which of the 3 objects is the smallest?
A. The object on the right is the smallest object.
B. The object on the left is the smallest object.
C. The object in the middle is the smallest object.

## Example rationale for MMC instance

**Step 1: Recognize the Type of Image**

- **Skill:** Perception: Visual Recognition, Image Classification, Graph identification
- **Evidence:** The image displays labeled axes, a plot area with data, and a legend/title indicating it is a chart.
- **Conclusion:** The image is a line chart.

**Step 2: Identify the Content of the Chart**

- **Skill:** Knowledge: Reading Comprehension, Understanding title and labels
- **Evidence:** The title reads "Participation rate in sports activities in Flanders (Belgium) from 2008 to 2018."
- **Conclusion:** The chart is about the participation rate in sports activities in Flanders over time.

**Step 3: Determine the Specific Year of Interest (2014)**

- **Skill:** Perception: Visual understanding, Locating points on the graph
- **Evidence:** On the x-axis, locate the year 2014.
- **Conclusion:** The point corresponding to 2014 is identified on the graph.

**Step 4: Compare the Participation Rate Around 2014**

- **Skill:** Perception: Visual Analysis, Comparing values over time
- **Evidence:** Observing the height of the line at 2014 compared to the previous year (2013) and the subsequent year (2015).
- **Conclusion:** The line shows a rise from 2013 to 2014.

**Step 5: Determine If There Was a Significant Increase**

- **Skill:** Reasoning: Logical Deduction, Identifying significant changes
- **Evidence:** The upward movement from 2013 to 2014, along with the slight downward/plateau from 2014 to 2015.
- **Conclusion:** There was a noticeable increase from 2013 to 2014.

' There was a significant increase in something specific in 2014.' Please answer whether the description is true or false according to the image.
A. True
B. False

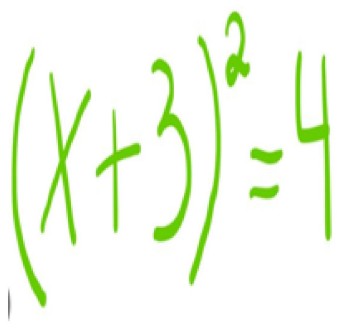

What is x in the equation?

### Example rationale for MMVet instance

**Step 1: Understand the Equation**

- **Skill:** Knowledge: Algebra, Equation understanding
- **Evidence:** The given equation is $(x + 3)^2 = 4$.
- **Conclusion:** We understand that we need to solve for $x$ by dealing with the squared term.

**Step 2: Take the Square Root of Both Sides**

- **Skill:** Reasoning: Algebraic manipulation, Solving equations
- **Evidence:** To simplify $(x + 3)^2 = 4$, we take the square root of both sides.
- **Conclusion:** $\sqrt{(x + 3)^2} = \pm\sqrt{4}$, which implies $x + 3 = \pm 2$.

**Step 3: Solve for $x$ in Both Cases (Positive and Negative Roots)**

- **Skill:** Reasoning: Algebraic manipulation, Solving linear equations
- **Evidence:**

  - $x + 3 = 2$
  - $x + 3 = -2$

- **Conclusion:** Solving each case:

  - $x + 3 = 2 \implies x = 2 - 3 = -1$
  - $x + 3 = -2 \implies x = -2 - 3 = -5$

ANSWER: $x = -1$ and $x = -5$.

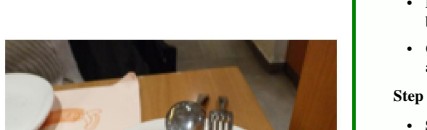

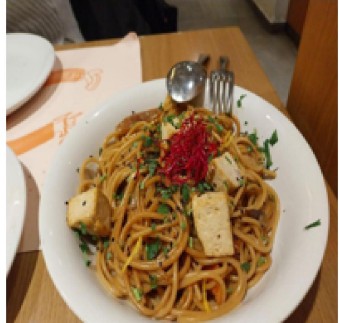

Is this safe for a vegan to eat?

### Example rationale for VibeEval instance

**Step 1: Identify Main Ingredients Visible**

- **Skill:** Perception: Visual Recognition, Food Identification
- **Evidence:** The image shows a dish with noodles, tofu, herbs, and possibly vegetables (visibly distinct green garnishes and red and yellow accents).
- **Conclusion:** The visible ingredients are noodles, tofu, herbs (possibly parsley or cilantro), and some vegetables.

**Step 2: Identify Potential Non-Vegan Ingredients**

- **Skill:** Knowledge: Food Composition, Typical Non-Vegan Ingredients in Pasta Dishes
- **Evidence:** Common non-vegan ingredients in pasta dishes include dairy (cheese, cream), meat, fish products (fish sauce), and eggs (in some types of noodles).
- **Conclusion:** There is no visible dairy, meat, or fish product in the dish. The presence of tofu suggests a plant-based protein source, but whether the noodles contain eggs cannot be determined just by sight.

**Step 3: Consider Potential Hidden Non-Vegan Ingredients**

- **Skill:** Knowledge: Vegan Diet Requirements, Awareness of Common Hidden Ingredients
- **Evidence:** Some sauces might contain non-vegan ingredients such as oyster sauce, fish sauce, honey, or certain seasonings.
- **Conclusion:** While tofu and the visible vegetables are vegan, the noodles or sauce might still contain non-vegan ingredients.

**Step 4: Conclude Based on Visual and Common Knowledge**

- **Skill:** Reasoning: Logical Deduction, Conclusion Drawing
- **Evidence:** Given the visible ingredients (tofu, vegetables, herbs) and absence of any obvious animal products, the dish appears vegan. However, without confirming the ingredients of the noodles and sauce, it's uncertain.
- **Conclusion:** Visually, this dish appears to be vegan-friendly, but it is important to confirm the ingredients of the noodles and sauce to ensure it is entirely safe for a vegan diet.

