# OpenReview forum: "Unearthing Skill-level Insights for Understanding Trade-offs of Foundation Models"
_ICLR.cc/2025/Conference — ICLR 2025 Poster_

### Official Review · Reviewer_AkzK · 2024-11-05

**Soundness:** 3
**Presentation:** 2
**Contribution:** 3
**Rating:** 6
**Confidence:** 4

**Summary:**

This paper proposes an automatic approach to infer underlying skills for evaluation instances by inspecting model-generated rationales. This helps in achieving a finer-grained understanding of model capabilities from existing benchmarks, increasing the interpretability of evaluating LLMs with benchmarks.

**Strengths:**

- The way of analyzing the model evaluation benchmark from the aspect of inferred skills is interesting and novel to me.

- This work utilizes a post-hoc verification of skills and validates its validity by comparing the results with human validation.

- The authors conduct experiments to show some interesting findings on skill generation and model evaluation.

**Weaknesses:**

Overall, the presentation needs improvement such as:
- A simple prompt example used to generate a detailed rationale shown in Figure 2 would be better.
- A clear subsection of used benchmarks and LLMs in the main paper.

I am confused about the description of the experimental section such as:

- Figure 8 is not clear enough at first sight. Which method uses skill annotations? And does the higher precision @ k = 20 indicate better performance or worse? Could you please explain the reason behind the evaluation?
- Figure 9 only depicts the number of skills per instance obtained by direct prompting vs. our rationale parsing. While the larger number cannot show the generation effectiveness. Reliability or relevance to the question are more important. Is there any other ablation study showing the comparisons concerning the reliability of annotated skills?

**Questions:**

See weaknesses.

---

> ### Author Response · Authors · 2024-11-21
> **Rationale-parsed skills have similar relevancy + result in more/larger slices; Examples + dset list added**
>
> Thank you for your time and valuable feedback. We highlight new analyses and updates to the draft:
> 1. Appendix C.1 has new quantitative analyses showing that our method results in equally relevant skills with much greater diversity in granularity as well as higher count (shown previously) than skills from direct prompting.
> 2. Appendix H has new examples of instances in our corpus along with generated skills and rationales; explicit references to this and other relevant appendices have been added in the main text. Full list of datasets will also be added shortly to the main text.
> 3. Figure 8 has been updated to revise a typo.
>
> We address concerns in detail below.
>
> ----
>
> **Deeper comparison of obtaining skills with direct prompting vs. our method.** We appreciate your suggestion to more closely compare skills obtained from our method (rationale parsing) vs. direct prompting. To this end, we have added two analyses in Appendix C.1, quantitatively supporting our previously anecdotal observations that rationale parsing leads to skills with (a) marginally lower relevance, and (b) much greater diversity in grain (i.e. both fine-grained and coarse-grained skills are included). Specifically, we utilize automatic post-hoc verification to measure the relevance of skills from direct prompting. The relevancy rate for skills obtained with direct prompting is 96.6%, which slightly surpasses the 94.1% we obtain for rationale-parsed skills. Then, we explore the granularity of skills listed by each method, which affects the sizes of slices that can be formed. In short, we observe a large fraction of skills inferred with direct prompting to be relevant for a very small number of instances, making them potentially too fine-grained to result in sufficiently large skill-slices.
>
> In summary, rationale parsing leads to a significantly higher count and diversity in grain of annotated skills than direct prompting, at no cost to relevancy of annotated skills. Furthermore, rationale parsing is necessary for our probing analysis, as we utilize rationale steps to form each probing question. We thank the reviewer for suggesting this deeper analysis, and we updated Appendix C.1 to reflect the new results.
>
> **Examples added for clarity.** We will include a full list of datasets in the main text (via fig. 3) in an updated draft, with explicit reference in the text, as well as greater reference to Appendices B and C, where a detailed list of datasets and the complete prompts are provided. We have also added more examples to Appendix H, showing what instances in our corpus, generated rationales, and annotated skills look like.
>
> **Clarifying figure 8.** We apologize for what was a typo in Figure 8 (thanks for helping us catch this!), where one bar was missing and thus labels were out of order – we have now updated it. Higher precision indicates better performance, as it reflects that a higher fraction of retrieved instances test the skill of interest (i.e. the query). There are two reasons for this experiment: (1) Retrieving instances based on the skills they test can be a very useful application of our work, as it will allow for the construction of custom evaluation sets or potentially even skill-specific training sets. (2) On this task, we can directly compare our skill slicing to prior methods for grouping instances to get richer evaluation output. The significantly improved skill-based retrieval of our method vs. embedding or attribute based methods show that our skill annotations offer novel signal over prior approaches.

---

> > ### Comment · Reviewer_AkzK · 2024-11-22
> > **Thanks for your response**
> >
> > Thanks for addressing my concerns. The additional comparison regarding skill acquisition is persuasive, and I will increase my rating.

---

> ### Author Response · Authors · 2024-11-27
>
> Thank you for reading our rebuttal! We are happy to see your concerns have been addressed. And we have updated fig 3 as requested to include all dataset names.

---

### Official Review · Reviewer_VpcF · 2024-11-08

**Soundness:** 3
**Presentation:** 3
**Contribution:** 3
**Rating:** 6
**Confidence:** 4

**Summary:**

This paper mainly studies the evaluation challenge from testing multiple skills in modern LLM evaluations. Given a specific benchmark dataset, the skill-wise performance is obscured and the possible hidden rich signal is usually underutilized. To alleviate this issue, this paper propose a automatic approach for recovering the underlying skills given any evaluation instance. In specific, this is implemented by inspecting model generated rationales. By examining the relevance of rationale-parsed skills and inferring skills for 46k instances over 12 benchmarks, it is shown that many skills to be common across benchmarks, resulting in the curation of hundreds of skill-slices. These skill slices are finally further shown to deliver novel insights on model trade-offs.

**Strengths:**

- First of all, the LLM evaluation problem is very important, especially under the case where multiple or many skills are evaluated together. This has been shown by Figure 1 as well to clearly clarify the motivation of this work.

- The paper is well written and easy to follow

- Automatic discovering the skills related to any evaluation instance is an important topic.

- Some findings are quite interesting. For instance, Gemini 1.5 Pro to be much stronger in math and science skills, like ‘computing molar mass’, while falling far behind for legal skills like ‘applying constitutional law’.

**Weaknesses:**

- The automatic verification protocol should be further explained. This is the single most important part of this method pipeline. From my understanding, querying LLM models or APIs with specific prompt engineerings can always get some sort of skills, rationales or chain of thoughts. The more important thing is how can you guarantee these generated content are properly validated. The author claimed that they will "release all our rationales, skill annotations, and skill-slices, which we term the Skill-Index, to the public.", but more explanation should be provided. The verified trustworthy accuracy here will be quite important.

- The technique contribution looks a bit limited. All the work done here are PE related work.

**Questions:**

- Assume in the industry, you are not allowed to access those strong models (e.g., GPT-4o). Given a specific LLM, the skill discovering ability is actually constrained by the skill ability of this LLM. How could you explain the effectiveness of automatic method, if let's say the model itself is weak in one category of skill while you expect this LLM can identify this skill and generate the corresponding rationales automatically?

- Given the generated rationales will be treated as a golden standard for the future use in other model evaluations, how to guarantee the trustworthiness of such rationales? This is actually the same question listed in weakness part.


- I am curious: How is the cost of conducting such kind of research?

---

> ### Author Response · Authors · 2024-11-21
> **explaination of multi-faceted validation and cost estimate**
>
> tldr of additions during rebuttal:
> - An extended **explanation of our validation**. We use a multi-faceted approach, leveraging human validation, post-hoc verification, inter-verifier agreement, and inter-annotator agreement, which all reflects a high level of skill accuracy.
> - **Cost estimate** of \\$1100 for 46,000 instances, or \\$0.02 per instance, cheaper than any human annotation. Also, this cost only needs to happen once – our annotations will be released for free.
> - **Skill annotation with Llama 3.2**; we find these skills to be highly reliable, suggesting closed-source models are not required.
>
> ----
>
> Thank you for the insightful feedback! We now address each concern.
>
> **Trustworthiness of annotated skills**. We concur that ensuring that annotated skills are relevant is crucial. As such, we devised multiple independent methods to validate skill annotations:
> 1. **Human verification**: we inspect 640 skills with a human judge, finding 95.7% of them to be relevant.
> 2. **Post-hoc verification: using 4 different judge models**, we automatically verify that annotated skills are indeed relevant. Importantly, we include irrelevant (i.e. randomly sampled negative) skills, so to ensure that the judge models do not simply say that all skills are relevant. We find that all judges overwhelmingly (a) mark annotated skills as relevant (94%), (b) mark irrelevant skills as irrelevant (91%)(c) agree with one another’s judgements (>90%, see figure 10). Point (a) shows our skills can be trusted, and points (b) and (c) show our verification scheme can be trusted. Also, 94% of human and automatic judgements agree, further validating our automatic verification procedure.
> 3. **Inter skill-annotator agreement**: in a more stringent assessment, we check to see if different rationale-generating models list similar skills. Even though skills from two models can differ while still both being relevant (i.e. if the two models solve the problem in distinct ways), we find a high overlap (>80% on average) for skills from 4 distinct annotators, adding more evidence to the trustworthiness of our skills.
> 4. **Corroboration with follow up analysis**: as reviewer uBHn points out, our latter experiments (routing, probing questions) offer indirect validation of our skill annotation, as insights are consistent with one another: probing questions reveal that models contradict themselves more frequently for skills where they achieve low slice accuracy, and routing experiments show the performance of a model on an unseen question is higher if the model is better at questions annotated with similar skills.
>
> The agreement across our numerous direct validation techniques offers strong evidence to the trustworthiness of our skill annotations. We add this extended explanation of our verification procedure to Appendix C.2.
>
> **Cost.** We note that skill annotation and verification only need to occur once per benchmark and will open source our annotations for 12 benchmarks. Thereby, we reduce the  cost to the community to use the vast set of annotations we already obtained. We estimate the **total cost for annotating verifying skills for the 46000 instances in our corpus to be less than \$1100** (breakdown below). Further, we find new promising evidence that open source-models can be used in place of closed ones. Specifically, during the rebuttal, we obtain skills using Llama 3.2V 90B as the rationale generating model for a subset of MMLU Pro and find Llama annotated skills to have 94% relevancy rate.
>
> Full breakdown of cost estimate.
> - Rationale generation: 60.7 tokens per question on average + 988 tokens for system prompt = 1048.7 input tokens per instance on average. Average response length is 377.5 tokens. The cost per input token is \\$5 per 1M and the cost per output token is \\$15 per 1M. There is also a \\$0.008 cost per API call. Thus, per instance, the cost is 1048.7 * 5 / 1M + 377.5 * 15 / 1M + 0.008, which comes out to \\$0.019 per question. Rounding this to \\$0.02 and multiplying by the 46,000 instances we annotate yields a cost of \\$920. For about 30k of these instances, we additionally pass an image, which costs 255 tokens / instance * \\$5 / 1M tokens * 30k instances  = \\$38 total. This yields a final total of \\$958 for all skill annotations.
> - Skill verification is dramatically cheaper, as the system prompt and (especially) the outputs are much shorter. Namely, the system prompt requires only 62 tokens and the average output length is about 130 tokens. This comes out to a cost of \\$0.002 per instance (1/10th the cost of rationale generation), which results in about \\$100 to do verification.
>
> We note that we used the May release of GPT-4o; the most recent release is notably cheaper (50% and 33% less for input and output tokens respectively). Batching inference can also bring down costs, as the (long) system prompt is shared across rationale-generating calls. We estimate token lengths with [1].
>
> (continued)

---

> > ### Author Response · Authors · 2024-11-21
> > **Evidence that open source models can also generate rationales + annotate skills**
> >
> > **Contributions.** We offer a novel approach to extract much richer signal from existing evaluations, resulting in actionable insights about the leading foundation models. These insights have direct impact, via (i) increasing downstream accuracy with skill-based routing, (ii) uncovering skill deficiencies common across models, and (iii) enabling skill-based data curation. Methodologically, we note our use of rationales as opposed to simple/direct skill prompting, which (in new rebuttal experiments) we find leads to a higher count and size of resultant skill-slices (Appendix C.1). In a second set of rebuttal experiments, we find our methodology is robust to the choice of rationale-generating model, being effective even for the open source Llama 3.2 model. Lastly, we devise numerous scalable validation approaches to (i) verify the motivating observation that models can articulate skills, even on instances they may not solve correctly, and (ii) allow for skill annotations for future datasets to be added to our corpus in a reliable manner.
> >
> > **What if we cannot use GPT-4o?** We find evidence that open-source models can provide trustworthy skill annotations, as Llama 3.2 generated skills on MMLU Pro obtain a 94% relevancy rate. To your latter question of how skills can be trusted if the annotating model itself struggles for a given question, we point to the right panel of Figure 3. Namely, we observe the relevance of listed skills is roughly equal, regardless of if the annotating model answered the question correctly. This suggests that LLMs are likely able to list relevant skills even if they ultimately apply the skill incorrectly, consistent with some notions from prior work showing that models may be useful to provide guidance for problems like complex planning even when they may not be able to fully complete the task on their own [2].
> >
> > [1] OpenAI Pricing Tool Calculator, https://gptforwork.com/tools/openai-chatgpt-api-pricing-calculator
> >
> > [2] Kambhampati et al, LLMs Can't Plan, But Can Help Planning in LLM-Modulo Frameworks, ICML ‘24
> >
> > We hope these comments help address your concerns! We would be more than happy to answer any follow up questions.

---

### Official Review · Reviewer_9gum · 2024-11-09

**Soundness:** 3
**Presentation:** 3
**Contribution:** 3
**Rating:** 8
**Confidence:** 3

**Summary:**

This paper proposes a new method to evaluate LLMs by analyzing their skills rather than just relying on overall accuracy scores. It extracts detailed reasoning steps (rationales) from models like GPT-4 to identify the skills used, grouping these into "skill-slices". By doing this, the researchers can reveal strengths and weaknesses that are not revealed through traditional evaluations. They also introduce probing questions to validate the identified skills, ensuring that the analysis reflects the model's true abilities. This approach aims to provide a deeper understanding of model capabilities and guide improvements for future models.

**Strengths:**

- The paper provides a new insight to evaluate LLMs by evaluating on fine-grained skill analysis rather than traditional scores. This method of extracting "skill-slices" from model-generated rationales allows for a deeper understanding of models’ capabilities. The insight is reasonable, and the meaningful for current LLM studies.
- The design of probing questions allows for a robust validation for the extracted skills, which could reduce the risk of hallucination.
- The paper provides extensive experimentations across multiple benchmarks and models. It clearly provides solid analysis and findings which might benefit for future studies.
- The paper provides skill annotations, which would guide future model and dataset developments.
- The skill-slices design could benefit for the explainability and transparency of large language models.

In general, it is a good paper.

**Weaknesses:**

- Real-world tasks often require a combination of multiple skills. The current method of skill extraction assumes that skills can be categorized and analyzed in isolation. This overlooks the fact that the interaction between multiple skills may be non-linear and context-dependent, meaning that combining skills does not always produce the same outcome as using those skills independently.
- Some LLMs may operate as "black boxes," where the true internal reasoning process is not fully transparent. This means that the skills inferred from the generated explanations may not align with the model's real decision-making processes.

The paper has already discussed some limitations.

**Questions:**

Given that models may hide internal process and not explicitly demonstrate all the skills they use, how would we evaluate those models?

---

> ### Author Response · Authors · 2024-11-21
>
> We thank the reviewer for their kind words and valuable feedback. We address some comments below.
>
> **Measuring challenge of skill combinations**. We completely agree that a model’s proficiency in applying skills independently may differ from its proficiency in using skills in combination – in fact, we make explicit mention of this at the end of section 3. We note that *skill annotations open the door for us to more closely measure these second order effects*. For example, instead of forming a skill-slice defined by a single skill, we can form slices defined by the presence of two skills at once. Then, we can compare performance on the slice defined by two skills to the performances on slices defined by each skill alone, or also the performance on probing questions for each skill separately. We leave this analysis out of scope of this paper, as methods for combining groups have been more closely studied in prior work [1], albeit not for skill slices. We hope our work enables further study of this phenomenon through the new lens of skill-combinations, as opposed to attribute combinations.
>
> **Rationales need not be faithful, so long as listed skills are still relevant to the underlying instance (which we verify)**. We also agree that generated rationales may not necessarily perfectly reflect a model’s internal processes. Thus, while annotated skills may not perfectly reflect what the rationale-generating model does, we empirically demonstrate that the annotated skills are still highly relevant to the given evaluation instance. That is, we do not use rationales to directly gain insight about the rationale-generating model, but instead, we use them to gain insight on evaluation data, with which we can later study any model. We make explicit note of this in the introduction, and add further mention of this in appendix C.

---

> > ### Comment · Reviewer_9gum · 2024-12-01
> >
> > Thanks for your response.

---

### Official Review · Reviewer_ubHn · 2024-11-12

**Soundness:** 3
**Presentation:** 3
**Contribution:** 3
**Rating:** 6
**Confidence:** 4

**Summary:**

This paper uses LLMs to annotate benchmark instances with skills required to solve them. The authors validate their annotation in multiple different (mostly automated) ways. They also analyze how different models' capabilities vary when restricted to so-called skill slices, i.e. sets of benchmark instances that all require a specific skill to solve (according to the automated annotation).

**Strengths:**

- The writing is generally quite engaging
- Decomposing broad benchmarks into more specific ones seems quite useful for things like model selection, whenever models are to be deployed in a specific setting.
- Despite the caveats listed in the weaknesses, the results on differences in model capabilities are quite interesting. The additional validation experiments, like the one on model routing, make me believe that there is indeed meaningful signal in the results.

**Weaknesses:**

- All results on comparing pairs of models would strongly benefit from adding error bars to show that the differences are indeed exceeding what would be expected from noise when comparing equally performant models.
- Additional statistical analysis to show that the most stark model differences cannot be explained by random noise + testing many hypotheses (i.e. comparing models on many different skills) would also strenghten the results.
- Releasing the annotations would be helpful for the reviewers to get a better intuitive picture of the validity of the proposed methods.

- Nitpicks:
   - The writing sometimes seems a bit too grandiose:
       - For example, whether the paper is "providing a valuable resource for the research community" seems like something the community will decide on.
  - The formatting in Appendix E is off (lots of empty space)
  - Appendix G is incomplete (it ends mid-sentence)
  - Some kind of face-value evaluation (such as a small scale human evaluation) beyond GPT-4 as judge would be useful for section 5.

**Questions:**

- What does figure 8 show? Is the last row's annotation a typo?
- What happens when combining skill and text/image embeddings in figure 8?
- What are the pros and cons of the verification approach based on negative samples?
- Which skills and benchmarks does the routing approach improve most on (let's say compared to an ensemble, as well as "perfect" routing that has access to the ground truth)?

---

> ### Author Response · Authors · 2024-11-21
> **Added markers of statistical significance and more example rationales**
>
> Thank you for your insightful comments and kind words. We address each point below.
>
> **Measures of statistical significance added.** We thank the reviewer for this valuable suggestion. We have added symbols to indicate statistical significance in the comparison of models. Namely, in figure 5, where strengths and weaknesses of each model are shown, we add a dagger symbol when the largest gap (best vs. worst model out of 3) is statistically significant and two daggers when both gaps are statistically significant (best vs. worst, and best vs. second), according to the Wilcoxon signed–rank test with a p-value threshold of 0.01. Of the 60 skill-wise strengths and weaknesses we show over three models, we find 47 of them to contain at least one statistically significant difference in accuracy between two models on the given skill-slice.
>
> **Example rationales and skills added.** Again, thank you for this very actionable suggestion! We have added examples of instances from many benchmarks, along with the generated rationale and annotated skills in Appendix H, supplementing the example presented in Figure 2.
>
> Answers to other questions:
> - **On including negative samples**: The pro to this is that we can ensure that the verifier model is not simply listing all skills as relevant – in other words, it serves as a unit test of sorts for our verifier model. The con is that the cost of verification is higher, though we note that the added cost is relatively small, as most of the tokens in each verification call come from the system prompt and the evaluation instance, not the skills that the verifier must check.
>
> - **The benchmarks that are improved most by routing** are those that aggregate questions testing many distinct skills, especially when those skills include ones where the models we study have different strengths. Namely, MMLU Pro and SEEDBench (our two largest and broadest benchmarks) see the largest gains by routing. We include more details and a full breakdown of routing performance in Appendix D.
>
> - **Figure 8 updated**: We apologize for what was a typo in Figure 8 where one bar was missing and thus labels were out of order – we have now updated it. This figure shows that skill annotations are very useful (and notably better than other annotations, like attributes or input embeddings) when attempting to retrieve instances based a query skill the instances should test. Combining skill annotations and embeddings is a very interesting idea! We leave it out of scope for now, though we conjecture that the two methods may be complementary and stronger when combined.
>
> - **Appendix spacing resolved**. We apologize for the formatting issue and incomplete sentence in the appendix. An old version of the draft was accidentally submitted in the final hour – hence the issue with Figure 8 as well – thank you for helping us catch this!

---

> > ### Comment · Reviewer_ubHn · 2024-11-22
> >
> > Thank you!
> >
> > What was the total number of skill pairs considered (The shown ones appear to be a subselection), and do your p-values account for multiple hypothesis testing?
> >
> > Regarding the routing results, it appears as if skill routing underperfoms GPT-4 on a few smaller benchmarks. Do you have a hypothesis for what is going on there?

---

> > > ### Author Response · Authors · 2024-11-27
> > >
> > > tldr: Yes, the level of significance we observe goes beyond noise from multiple hypotheses -- longer discussion below. And for routing, qualitatively, the difficulty for some skills between datasets can vary at times, which may contribute to the slightly lower routing performance. It would be interesting to see if we could supplement skill annotations with other information about an instance in a future iteration of routing.
> > >
> > > Re multiple hypotheses: The total number of pairs is 332 (num skill slices) * 3 (num pairs of the 3 models) = 996. At a p-value of 0.01, we'd expect roughly 10 significant pairs due to noise, which is far less than the (at least) 47 we observe in the subset of comparisons we inspect. Moreover, this 47 count is not only a lower bound for what is in the plot, but also a much lower bound overall -- in practice, we find many more significant skill-wise differences that we exclude for brevity.
> > >
> > > Thanks again for all your engagement and feedback!

---

> > > > ### Comment · Reviewer_ubHn · 2024-11-28
> > > >
> > > > Thank you for the reply!
> > > >
> > > > My concerns about the lack of statistical analysis have largely been addressed (even though I disagree with some details like not directly employing a correction for multiple hypothesis testing).
> > > >
> > > > I will keep my score at 6 for multiple reasons:
> > > > -  I am generally reluctant to give out a score of 8, and there are no intermediate options between 6 and 8
> > > > -  While most of my concerns have been addressed to some extent, they are often only resolved about 80%. For example, annotation examples were only made available for a subset of the considered benchmarks, and it is unclear how the examples were selected.
> > > > -  While I find the presented approach quite useful, I do worry that it could sometimes produce misleading results in practice for subtle reasons that are difficult to test for.

---

> > > > > ### Author Response · Authors · 2024-11-28
> > > > >
> > > > > Thank you for your careful consideration and continued feedback! We are glad that you still feel our submission should be accepted, and that it seems like you would boost us to a 7 if that was possible ;)
> > > > >
> > > > > Just to clear up your last couple comments for your ease of mind / in case anyone else had similar thoughts:
> > > > > - The lack of an example for 4 out of 12 benchmarks was purely cosmetic -- the image for the instance per excluded benchmark looked weird when resized, and we did not prioritize updating our visualization code as all rationales will be made available very soon and time was tight across rebuttals from different papers. We figured showing examples from 8 benchmarks would suffice in giving readers a robust taste of what rationales and annotated skills looked like.
> > > > > - Users can flexibly select the minimum size of skill-slice to inspect so to determine the level of uncertainty they are willing to tolerate. Insights can also be further validated by comparing outputs across similar skill-slices (e.g. the consistently poor relative performance of Gemini across distinct legal skills) or utilizing our probing question protocol.

---

### Public Comment · ~V_S_Silpa1 · 2025-03-15
**Code url not working**

Thanks for this valuable piece of contribution to the research community. Currently, the github link given in the paper is broken, It would be great if the authors could fix this as early as possible or provide an estimated timeline in which this would be fixed since the primary contribution of this paper is Skill-Index.

---

> ### Public Comment · ~Mazda_Moayeri1 · 2025-03-20
>
> Hi, thanks for the kind words. Try again now -- the github link was stuck on private. Feel free to directly email me with follow up questions :)

---

### Meta-Review · Area_Chair_WziY · 2024-12-21

**Metareview:**

(a) The paper proposes an automatic approach to evaluate LLMs by analyzing skills through model-generated rationales, rather than relying solely on aggregate accuracy. The method extracts skills from rationales for 46k instances across 12 benchmarks, creating "skill-slices" that test common abilities. The key findings show that models like GPT-4o, Claude 3.5 Sonnet, and Gemini 1.5 Pro have distinct skill-level strengths despite similar overall accuracy. For example, Gemini excels at computing molar mass but lags in constitutional law. The authors validate their approach through multiple methods and demonstrate practical utility through skill-based routing that improves accuracy.

(b) Strengths:
- Novel and well-validated methodology for extracting fine-grained skill information from existing benchmarks
- Comprehensive empirical validation through multiple approaches including human verification, automated checks, and inter-annotator agreement
- Strong practical impact demonstrated through routing experiments and model comparison insights
- Cost-effective approach ($0.02 per instance) that can be applied to future benchmarks
- The findings offer actionable insights for model selection and improvement

(c) Weaknesses:
- Could benefit from more analysis of how skills interact when combined
- Statistical significance testing could be more rigorous with explicit correction for multiple hypothesis testing
- Some benchmarks lack example annotations in the appendix (though this appears to be just a visualization issue)

(d) Accept: The paper presents a novel, well-validated approach that provides valuable insights into model capabilities at a granular level. The extensive empirical validation, practical utility demonstrated through routing experiments, and relatively low cost of implementation make this a significant contribution to the field. The weaknesses are relatively minor and many were addressed during the rebuttal period.

**Additional Comments On Reviewer Discussion:**

The reviewers raised several key points during discussion: statistical significance of model comparisons, validation of skill annotations, and cost/practicality concerns. The authors addressed these thoroughly by: adding statistical significance markers showing 47 of 60 skill-wise comparisons had significant differences; providing detailed validation through multiple approaches including human verification (95.7% relevance), automated checks (94% relevance), and inter-annotator agreement (>80% overlap); and demonstrating low cost ($0.02/instance) with evidence that open-source models like Llama 3.2 can also generate reliable skills. They also clarified figure formatting issues and added more example annotations. All reviewers were satisfied with the responses, with two explicitly increasing their scores. The remaining concerns about multiple hypothesis testing and skill interaction analysis were acknowledged as future work opportunities that don't detract from the paper's core contributions.

---

### Decision · Program_Chairs · 2025-01-22

Accept (Poster)